# HIV-1 Env trimers asymmetrically engage CD4 receptors in membranes

Wenwei Li[1✉], Zhuan Qin[1], Elizabeth Nand[1], Michael W. Grunst[1], Jonathan R. Grover[1], Julian W. Bess Jr[2], Jeffrey D. Lifson[2], Michael B. Zwick[3], Hemant D. Tagare[4], Pradeep D. Uchil[1] & Walther Mothes[1✉]

Human immunodeficiency virus 1 (HIV-1) infection is initiated by binding of the viral envelope glycoprotein (Env) to the cell-surface receptor CD4[1–4]. Although high-resolution structures of Env in a complex with the soluble domains of CD4 have been determined, the binding process is less understood in native membranes[5–13]. Here we used cryo-electron tomography to monitor Env–CD4 interactions at the membrane–membrane interfaces formed between HIV-1 and CD4-presenting virus-like particles. Env–CD4 complexes organized into clusters and rings, bringing the opposing membranes closer together. Env–CD4 clustering was dependent on capsid maturation. Subtomogram averaging and classification revealed that Env bound to one, two and finally three CD4 molecules, after which Env adopted an open state. Our data indicate that asymmetric HIV-1 Env trimers bound to one and two CD4 molecules are detectable intermediates during virus binding to host cell membranes, which probably has consequences for antibody-mediated immune responses and vaccine immunogen design.

HIV-1 begins infection of CD4[+] T cells when the trimeric HIV-1 Env binds to the cell-surface receptor CD4[1–4]. CD4 binding induces conformational changes within the gp120 subunit of Env that enable subsequent engagement of the co-receptors CCR5 or CXCR4. Receptor and co-receptor engagement trigger conformational changes in the Env gp41 subunits, driving fusion of the virus and host cell membranes. Structural insights into HIV-1 Env interactions with soluble CD4 (D1D2 domains) were gained initially with a gp120 core[5], and then with stable, soluble Env trimers (trimers containing an S–S disulfide bridge and an I559P mutation, or SOSIPs)[6–9]. In the closed state, the variable loops 1 and 2 (V1 and V2) form the apex of the Env trimer. CD4 binding results in an approximately 40 Å displacement of the V1V2 loop that aligns with the CD4 D1D2 domains[10–13]. The insights gained from soluble trimers have been confirmed with full-length Env proteins solubilized from membranes using detergent[14–16]. Further information about the co-receptor-binding step of Env was obtained by embedding the coreceptor CCR5 in lipid nanodiscs and determining the structure of the complex of soluble gp120, soluble CD4 (D1–D4) and CCR5[17]. Binding of CCR5 did not induce additional allosteric changes in gp120 but, rather, brought the CD4-bound gp120 closer to the lipid bilayer mimicked by the nanodisc. Although completely open and partially open CD4-bound Env conformations have been observed with SOSIP trimers[10,11], whether the partially open Env trimer conformations are true intermediates is unclear.

Further insights into Env–CD4 interactions have been gained using HIV-1 Env trimers in native membranes interacting with soluble receptors. Directly imaging virus particles using cryo-electron tomography (cryo-ET) enabled the characterization of the native Env trimer and the Env trimer opened by soluble CD4 initially at a resolution of about 20 Å (ref. 18) and more recently at around 9–10 Å (refs. 19,20). These structures largely confirm the high-resolution Env structures obtained with soluble trimers and soluble ligands[19]. Insights into the behaviour of HIV-1 Env molecules on the surface of virus particles have been gained from single-molecule fluorescence resonance energy transfer (smFRET) analysis, which indicated that individual gp120 protomers are dynamic and have spontaneous access to open conformational states, including the CD4-bound state[21,22]. Engineering trimers that can bind to only one or two CD4 molecules suggested that a necessary intermediate FRET state in the opening of Env corresponds to an asymmetric trimer in which only one CD4 molecule binds to the trimer[22]. Moreover, a structure of a soluble trimer mutationally prevented from opening has been observed to bind to only a single CD4[23]. Finally, the interaction of HIV-1 Env molecules with CD4 and coreceptor molecules has been studied in living cells by combining super-resolution localization microscopy with fluorescence fluctuation spectroscopy imaging[24]. These data suggested that HIV-1 entry is initiated by Env binding to a single CD4, followed by recruitment of additional CD4 molecules and a dimer of coreceptor molecules.

The interaction between HIV-1 Env trimers and CD4 molecules has not yet been structurally characterized in native membranes. Here we directly visualize the interactions between HIV-1 Env in virions and native membrane-bound CD4 by cryo-ET. We observed that Env–CD4 complexes cluster and organize into rings, and the patterns of clustering correlate with decreasing distances between membranes at the interfaces. Subtomogram averaging and classification revealed that, when the membranes were further apart, an Env trimer engaged

[1]Department of Microbial Pathogenesis, Yale University School of Medicine, New Haven, CT, USA. [2]AIDS and Cancer Virus Program, Frederick National Laboratory for Cancer Research, Frederick, MD, USA. [3]Department of Immunology and Microbiology, The Scripps Research Institute, La Jolla, CA, USA. [4]Department of Radiology and Biomedical Imaging, Yale University, New Haven, CT, USA. ✉e-mail: wenwei.li@yale.edu; walther.mothes@yale.edu

a single CD4 molecule. As the opposing membranes approached each other, Env trimers bound to two or three CD4 molecules. The V1V2 loop projected outward in the CD4-bound protomers, while the unbound protomers showed heterogenous conformational states. These data indicate that asymmetric HIV-1 Env trimers with one and two bound CD4 molecules are detectable intermediates during virus binding to membranes.

## Env–CD4 interactions imaged in membranes

We used cryo-ET to characterize membrane-embedded HIV-1 Env–CD4 interactions in situ using HIV-1BaL particles produced from chronically infected SUP-T1 T cells, which have a high density of Env trimers making them suitable for use in cryo-ET studies[18,19]. HIV-1BaL is a tier 1b primary isolate[25]. Consistent with an increased sensitivity of tier 1b isolates to CD4, we observed moderate shedding of gp120 into the supernatant in the presence of soluble CD4 (Extended Data Fig. 1). As these virus preparations are highly infectious, they were inactivated with aldrithiol-2 (AT-2) before imaging (Methods); AT-2 inactivation does not interfere with the CD4 binding or fusion activity of these particles[20,26]. To study HIV-1 Env interactions with CD4 receptor residing in biological membranes, HIV-1BaL particles were mixed with plasma membrane blebs generated from TZM-bl cells expressing CD4 receptor with CCR5 and CXCR4 coreceptors, and plunge-frozen for cryo-ET imaging. In the reconstituted cryo-tomograms, membrane–membrane interfaces were observed between HIV-1BaL particles and plasma membrane blebs (Fig. 1a,b). At these interfaces, multiple Env trimers formed clusters (Fig. 1c,e), and full-length CD4 was observed to interact with Env (Fig. 1d,f). Although we observed Env–CD4 interactions between HIV-1BaL and plasma membrane blebs, there were limitations within this system. The plasma membrane blebs varied in size (compare Fig. 1a versus 1b) and, as cryo-ET imaging is constrained by the depth of the vitreous ice layer (~500 nm), the image quality notably decreased with larger blebs. Furthermore, a high number of Env–CD4 complexes is required to obtain subtomogram-averaged structures. We therefore developed an experimental system that is uniform and presents observable Env–CD4 complexes at a high frequency by using viral particles, which are the ideal size for cryo-ET imaging (~150 nm). In this system, we used murine leukaemia virus (MLV) GagPol to produce virus-like particles (VLPs) that carry the CD4 receptor with or without co-receptor CCR5 (MLV-CD4 particles). When HIV-1BaL particles and MLV-CD4 particles were incubated, no gp120 shedding into the supernatant was observed (Extended Data Fig. 1). Mixtures of HIV-1BaL and MLV-CD4 particles were plunge-frozen for imaging by cryo-ET. HIV-1 and MLV have distinct capsid morphologies, enabling the identification of membrane–membrane interfaces that were the consequence of Env–CD4 interactions (Fig. 1g–i). Although individual Env molecules could be visualized on HIV-1BaL particles, the smaller CD4 molecules were not easily identifiable on MLV particles. However, after binding to Env at membrane–membrane interfaces, the CD4 receptor molecules could be visualized (Fig. 1j,k).

## Env clustering is induced by CD4 binding

Patterns of Env clustering and ring formation became apparent in the cryo-ET tomograms. Quantification revealed that the number of Env trimers increased from small clusters to large clusters, and to a slightly lesser extent in rings (Fig. 2a–e). Furthermore, the distance between the opposing membranes changed with small clusters having the greatest membrane distance, large clusters having an intermediate distance and rings having the smallest distance (Fig. 2f). We further quantified Env clustering by averaging membrane–membrane interfaces and overlaying all Env–CD4 complex coordinates (Fig. 2g). The unbiased analysis of all datapoints confirms that increasing Env clustering and organization into rings coincides with reduced membrane distance.

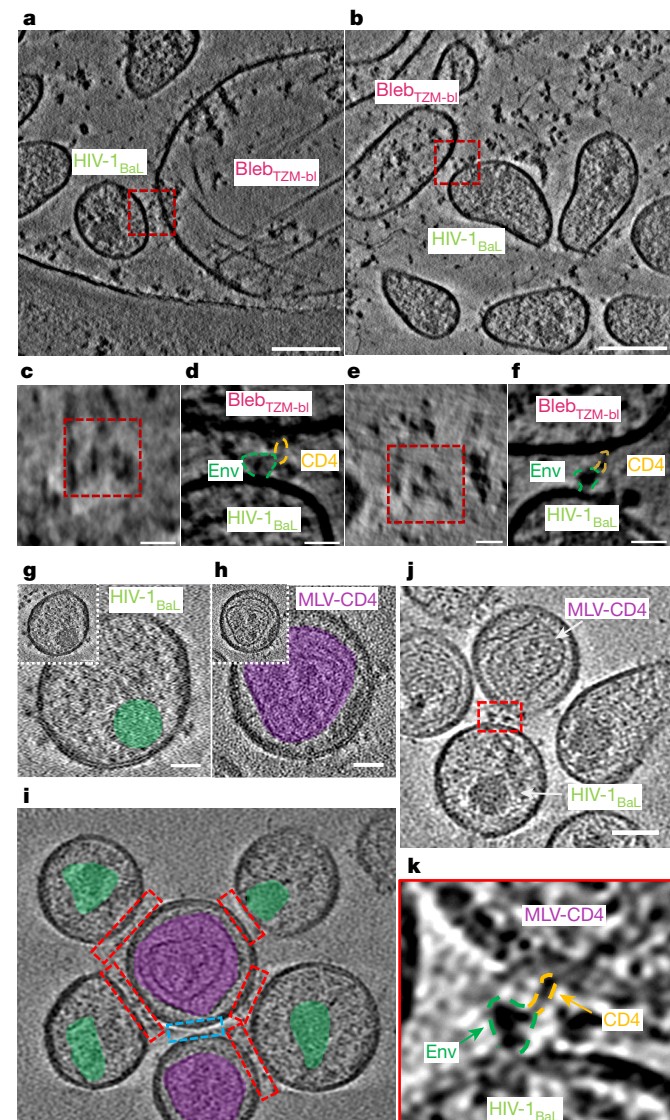

**Fig. 1 | Env–CD4 interactions are captured in biological membranes using cryo-ET. a,b,** Representative cryo-tomograms of membrane–membrane interfaces (red boxes) between HIV-1BaL viral particles and plasma membrane blebs generated from TZM-bl cells (blebTZM-bl). Scale bars, 100 nm. **c,e,** Top-down views of the membrane–membrane interfaces shown in **a** (**c**) and **b** (**e**), revealing Env clustering. Scale bars, 20 nm. **d,f,** Magnified images of the membrane–membrane interfaces shown in **a** (**d**) and **b** (**f**) depict Env–CD4 interactions in raw tomograms. Scale bars, 20 nm. **g,** Representative image of HIV-1BaL virus with Env on its surface. The capsid is highlighted in green. HIV-1BaL was inactivated with AT-2, resulting in less electron-dense capsids. Scale bars, 25 nm. **h,** Representative images of an MLV VLP carrying CD4 on its surface. The MLV capsid is highlighted in purple. Scale bars, 25 nm. **i,** A representative image of membrane–membrane interfaces (red boxes) in cryo-tomograms. Different capsid structures enable the identification of membrane–membrane interfaces as opposed to interfaces that do not have Env–CD4 interactions (blue box). Scale bar, 50 nm. **j,** A representative tomogram with the membrane–membrane interface highlighted in red. Scale bar, 50 nm. **k,** Magnified image of the membrane–membrane interface shown in **j**. Env–CD4 interactions are visible in raw tomograms. Scale bar, 10 nm.

To test whether the presence of CD4 correlated with Env clustering, all arc distances between two Env trimers were calculated in the presence and absence of MLV-CD4 particles. The presence of MLV-CD4 VLPs increased the frequency of short arc distances between Env trimers on the surface of HIV-1 virions (Fig. 2h). Likewise, a spatial cluster analysis

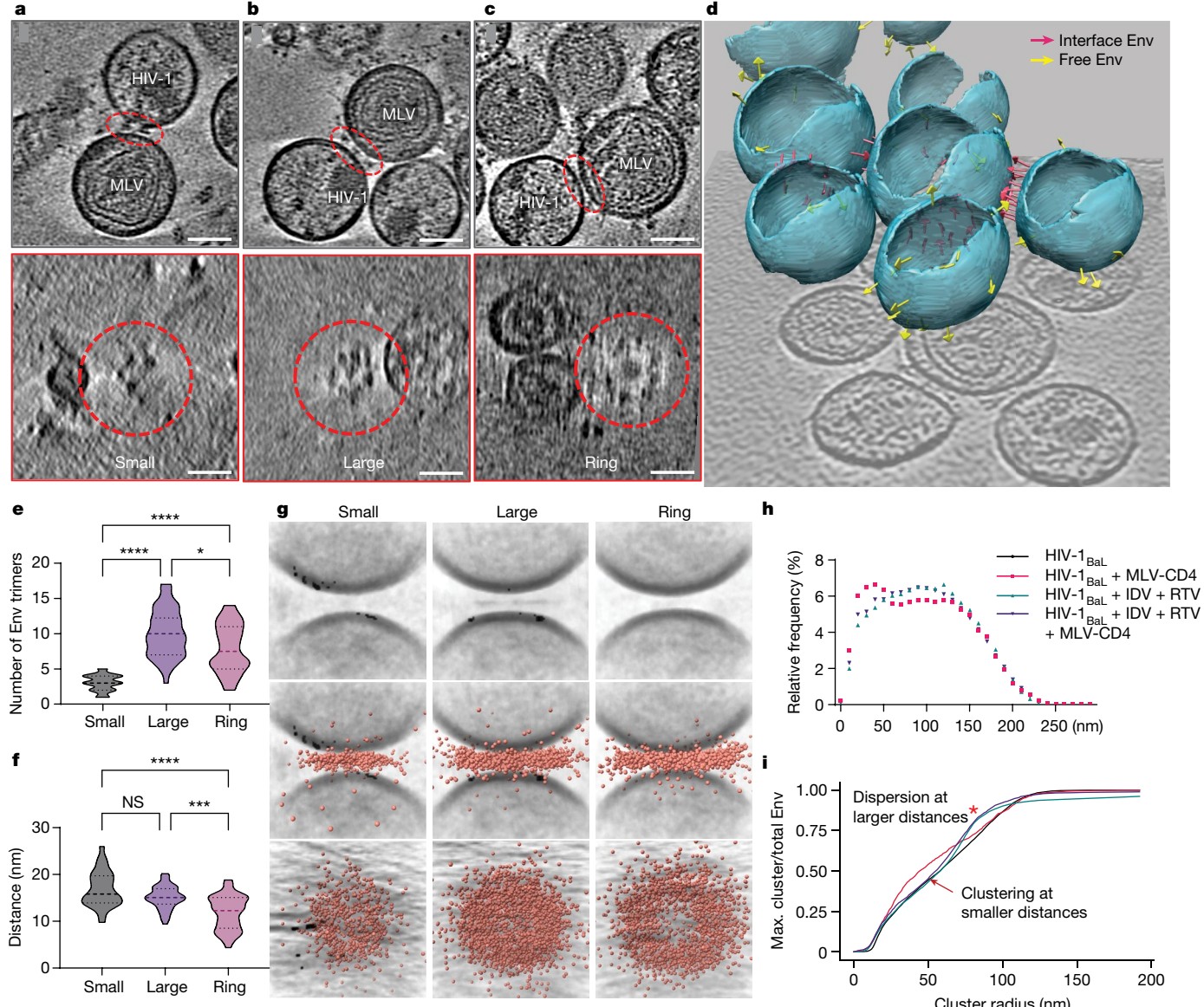

**Fig. 2 | HIV-1 Env binding to CD4 induces Env clustering and ring formation at membrane–membrane interfaces. a–c**, Representative tomograms of small (**a**), large (**b**) and ring (**c**) Env cluster formations. The interfaces are indicated by red dashed ovals. Bottom, top-down views of each interface, revealing Env clustering. Scale bars, 50 nm. **d**, Three-dimensional representation of viral particles in a representative tomogram. The pink arrows represent Env trimers at membrane–membrane interfaces. The yellow arrows represent free Env trimers on the surface of HIV-1$_{BaL}$. **e**, The number of Env trimers present at interfaces in each clustering pattern. Mean ± s.d. = 3.0 ± 1.1 (small), 10.0 ± 3.4 (large) and 7.9 ± 3.4 (ring). *$P$ = 0.0478, ****$P$ < 0.0001. **f**, The distance between the membranes at the interfaces for each clustering pattern. Mean ± s.d. = 16.9 ± 3.8 nm (small), 15.1 ± 2.4 nm (large) and 11.9 ± 3.6 nm (ring). NS, $P$ = 0.4178; ***$P$ = 0.0002, ****$P$ < 0.0001. **g**, Subtomogram-averaged interfaces from small,

large and ring clusters. Individual subtomograms of the membrane–membrane interfaces from each class were aligned, and the coordinates of Env–CD4 complexes were overlaid and displayed as side and top-down views (middle and bottom, respectively). **h,i**, Clustering analysis of Env trimers on the surface of mature (black) and immature HIV-1$_{BaL}$ particles alone (green, prepared by treating virus-producing cells with the protease inhibitors indinavir (IDV) and ritonavir (RTV)), and mature (red) and immature (purple) HIV-1$_{BaL}$ particles mixed with MLV-CD4 VLPs. **h**, Histogram profile of the arc distances between Env trimers on the surface of particles. **i**, Multidistance spacial cluster analysis of the ratio of the largest number of Env trimers with increasing cluster radii to the total Env trimers on each viral particle. Env clustering (arrow) and dispersion (red asterisk) are indicated. Max., maximum.

of Env trimers on HIV-1$_{BaL}$ particles pointed to an increase in the number of Env molecules accumulating at shorter radii in the presence of CD4-bearing particles (Fig. 2i). These analyses indicate that a sub-population of trimers is drawn into clusters at membrane–membrane interfaces in a CD4-dependent manner.

HIV-1 Env mobility on the surface of the virus particle has been observed to be dependent on capsid maturation and the cytoplasmic tail (C-tail) of Env[27–32]. In the immature capsid, Env is laterally immobilized by the C-tail tethering to the underlying immature matrix. During

maturation, the matrix protein is cleaved from the capsid, enabling Env to move laterally on the surface of the viral particle[28,29]. To test whether Env clustering induced by CD4 binding is impaired in immature capsids, MLV-CD4 particles were incubated with HIV-1$_{BaL}$ immature particles generated by treating virus-producing cells with protease inhibitors, then processed for imaging using cryo-ET (Extended Data Fig. 2a,b). Quantification of tomograms revealed that membrane–membrane interfaces with immature capsids had fewer Env trimers (4.6 ± 2.5, mean ± s.d.) than interfaces with mature capsids (7.5 ± 4) (Extended Data Fig. 2c). The

membrane distances at these interfaces showed only a minor change between mature and immature capsids (Extended Data Fig. 2d). Furthermore, the addition of MLV-CD4 particles to immature HIV-1$_{BaL}$ particles neither decreased the arc distances between two Env trimers (Fig. 2h) nor induced Env clustering at small radii in the spatial cluster analysis (Fig. 2i). These observations indicate that the CD4-induced clustering of Env at membrane–membrane interfaces is dependent on capsid maturation. Capsid maturation has been observed to allow for Env clustering on free virus particles[28–30]. Notably, capsid maturation had an effect on Env clustering in our experimental system, as Env on immature particles was nearly evenly dispersed at large cluster radii (Fig. 2i).

As a control, we used VLPs with high numbers of Env trimers. These VLPs were produced from a cell line that endogenously expresses HIV-1$_{ADA.CM}$ Env that has lost a portion of its cytoplasmic tail (Env(ADA. CM.755*), truncated at residue 755)[33]. We produced VLPs carrying Env$_{ADA.CM.755*}$ with either a wild-type HIV-1 *gag-pol* gene or HIV-1 *gag-pol* with a protease (PR)-knockout mutation that precluded cleavage of the Gag precursor. We imaged these VLPs on their own and in the presence of MLV-CD4 VLPs to test whether Env clustering was still dependent on capsid maturation even with a very high number of Env trimers on the surface of the virion. Given the very high trimer density, a high number of Env trimers assembled at membrane–membrane interfaces. Furthermore, as Env$_{ADA.CM.755*}$ has a truncated cytoplasmic tail, the clustering was independent of PR-dependent capsid maturation (Extended Data Fig. 3a–d). HIV-1$_{ADA.CM.755*}$ Env particles also provided an additional source of trimers to study the structure of Env–CD4 complexes at membrane–membrane interfaces.

## Env binds to one and two CD4 molecules

Approximately 5,700 subtomograms containing Env–CD4 complexes were manually picked from 168 tomograms. Subtomogram averaging of these particles generated a density map of the HIV-1 Env trimer bound to three native, membrane-bound CD4 receptor molecules at a resolution of around 15 Å (Fig. 3a–d and Extended Data Fig. 4a). The trimers were clearly open with the gp120 density positioned away from the central axis. The density for V1V2 loops projected outward in all three bound protomers of the Env trimer, which is consistent with previous high-resolution structures of soluble Env trimer bound to CD4[10,11] (Fig. 3b,d).

In this structure, one bound CD4 molecule featured stronger density than the other two, which had the lowest resolution as shown by local-resolution analysis. These differences indicate that there is structural heterogeneity in CD4 binding (Fig. 3d and Extended Data Fig. 4a). This, along with the above observations that the membrane distance varied among different patterns of Env clustering, suggested that the structural heterogeneity was linked to the distance between membranes. We therefore performed subtomogram classification on the basis of membrane distance, which revealed two subclasses: one with membranes further apart (-190 Å) whereby Env engaged only a single CD4 molecule (Fig. 3e–g and Extended Data Fig. 4b), and one with membranes closer together (-140 Å) whereby Env engaged three CD4 molecules (Fig. 3h,i). Heterogeneity remained in the densities of V1V2 loops and CD4 in the latter structure, so we performed further subtomogram classification on the basis of CD4 binding. This revealed an additional subclass of Env trimers with two bound CD4 molecules (Fig. 3j–m and Extended Data Fig. 4c,d). A total of 41% of Env trimers was bound to one CD4, 25% to two CD4 and 34% to three CD4 molecules (Fig. 3n). Thus, structures of Env bound to one, two and three CD4 molecules are observable intermediates at membrane–membrane interfaces (Fig. 3g,k,m and Extended Data Fig. 4).

As a control, approximately 6,000 unliganded Env trimers on HIV-1$_{BaL}$ particles from the same datasets were aligned and averaged. The resulting averaged structure achieved subnanometre resolution after $C_3$ symmetry expansion (Extended Data Fig. 4e). It revealed a closed conformation of the Env trimer (Extended Data Fig. 6d,h) consistent

with previous results[19]. The higher-resolution unliganded Env structure from the same dataset indicates that heterogeneity of Env–CD4 interactions limits higher-resolution determination. Superimposing the coordinates of Env bound to different numbers of CD4 molecules to the averaged interface revealed that Env trimers bound to two or three CD4 were concentrated towards the centre, whereas Env trimers bound to one CD4 were dispersed towards to the periphery (Fig. 3o and Extended Data Fig. 5a). These findings also aligned with the observation of Env–CD4 interactions in raw tomograms, in which Env trimers bound to one CD4 molecule were more frequently found in the peripheral region of the interfaces (Extended Data Fig. 5b), whereas Env trimers bound to two and three CD4 molecules were predominantly observed in the centre (Extended Data Fig. 5c). These distribution patterns further support the findings from the averaged subclasses, with the two membranes being further apart at the periphery (where Env binds to one CD4 molecule) and closer together at the centre of the interface (where Env binds to two and three CD4 molecules).

## Changes in CD4 support Env binding

We were able to resolve the structure of full-length, membrane-bound CD4 in the structure of Env bound to three CD4 molecules with a shorter membrane distance of around 140 Å (Fig. 3m). In addition to the familiar structure of the D1D2 domains bound to Env, the D3D4 domains became clearly visible. The D3D4 domains of all three bound CD4 molecules aligned along the target membrane. The observed angle between the D1D2 domains and the D3D4 domains is consistent with the previously observed flexibility in the D2–D3 hinge[34]. The alignment of the D3D4 domains and the flexibility of the D2–D3 hinge enabled CD4-bound Env to approach the target membrane. In the Env structure bound to one CD4 molecule, only the D1D2 domains of CD4 were resolved (Fig. 3g). To account for the observed membrane distance of about 190 Å, CD4 must be in a fully extended conformation to be able to reach Env on the opposing membrane. Thus, our data support a model in which conformational changes in CD4 facilitate HIV-1 binding to membranes.

## Env trimers asymmetrically engage CD4

All of the structures of Env–CD4 complexes had well-defined densities of the bound CD4, with each Env protomer engaging CD4 displaying adjacent density for the outwardly projecting V1V2 loops (Fig. 3f,j,l). All Env–CD4 complexes, regardless of CD4 binding, were missing EM densities at the apex region, suggesting an open conformation in which gp120 density moves away from the central axis (Extended Data Fig. 6a–d). This structure is similar to open conformations previously revealed by SOSIP trimers bound to soluble domains of CD4[10,11]. Notably, the free protomer in the trimer bound to two CD4 molecules lacked the density for the outwardly projected V1V2 loops. Although the resolution of our cryo-ET maps is not high enough to determine with certainty where the V1V2 loops sit, the conformational state is clearly distinct from the CD4-bound conformation (Fig. 3j and Extended Data Fig. 6f). Similar distinct conformational states are seen in the unliganded protomers of the Env trimer bound to one CD4 molecule, although some weak density for outwardly projected V1V2 loops remains visible (Extended Data Fig. 6e). This is probably attributable to remaining heterogeneity in the sample. This analysis indicates that the HIV-1 Env trimers bound to one and two CD4 molecules are asymmetric open trimers with the CD4-bound protomer adopting an outward rearrangement of the V1V2 loops, while the unliganded protomers remain in a distinct conformational state (Fig. 3f,j and Extended Data Fig. 6a,b,e,f).

## Env bound to three CD4s is partially open

Previous studies of CD4-D1D2-bound soluble trimers include B41. SOSIP.664 (Protein Data Bank (PDB): 5VN3) and BG505.SOSIP.664

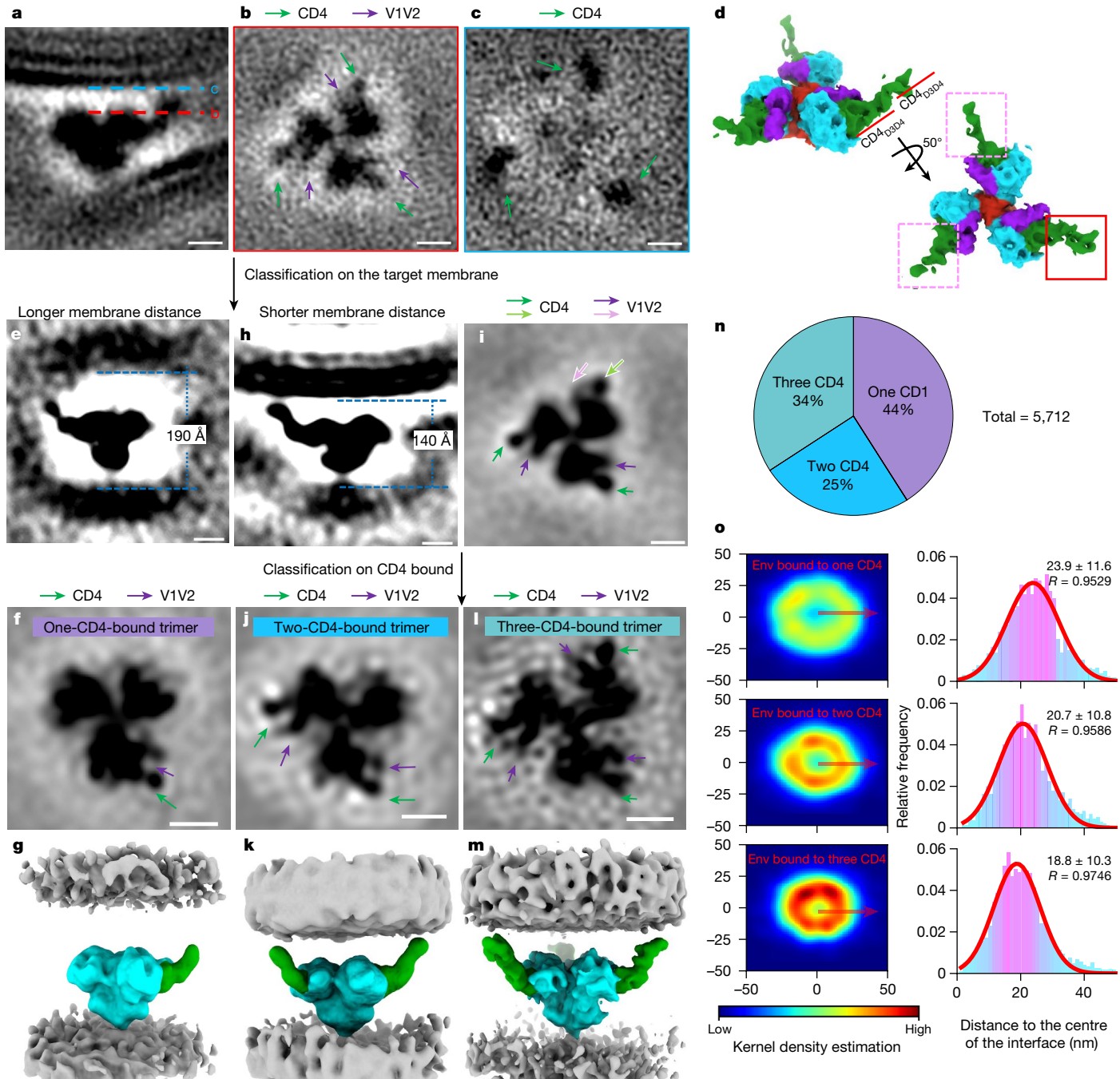

**Fig. 3 | Subtomogram averaging and classification reveal intermediates with HIV-1 Env trimers bound to one, two and three CD4 receptor molecules.** **a–c**, Side view (**a**) of the subtomogram average of the Env–CD4 complex at the membrane–membrane interfaces. The dotted lines indicate the positions of the top-down view slices in **b** (red) and **c** (blue). The densities of CD4 and V1V2 loops are labelled in green and purple, respectively. **d**, Segmentation of the Env–CD4 average structure. Side view (top) and top view (bottom) of Env (cyan) bound by three CD4 molecules (green). V1V2 loops are shown in purple and gp41 is shown in red. The CD4 molecule with strong density is indicated by a red box; the two CD4 molecules with weaker densities are indicated by pink boxes. **e–g**, Subclass average for longer membrane distance (190 Å) after focused classification on the target membrane. Side view (**e**), top view (**f**) and segmentation (**g**) are shown. **h,i**, Subclass average for shorter membrane distance (140 Å) after focused classification on the target membrane. Side view (**h**) and top view (**i**) are shown. The distinctive V1V2 loop and CD4 densities are indicated by pink and light green arrows, respectively. **j–m**, Focused classification on CD4 binding was performed in the subclass with a shorter membrane distance (**h,i**). Subclass averages of Env bound to two CD4 (**j,k**) and three CD4 molecules (**l,m**) are shown as top views (**j,l**) and segmentations (**k,m**). **n**, The proportion of Env trimers bound to one, two or three CD4 molecules. **o**, Distribution analysis of Env bound to one, two or three CD4 molecules at the interfaces. The kernel density heat map shows the 2D distribution of Env from the interface centre. The histograms show Env distances to the interface centre fitted with Gaussian curves (mean and *R* values are indicated). In the segmentations, Env is shown in cyan and CD4 is shown in green. Membranes are shown in grey. For **a–c**, **e**, **f**, **h–j** and **l**, scale bars, 5 nm.

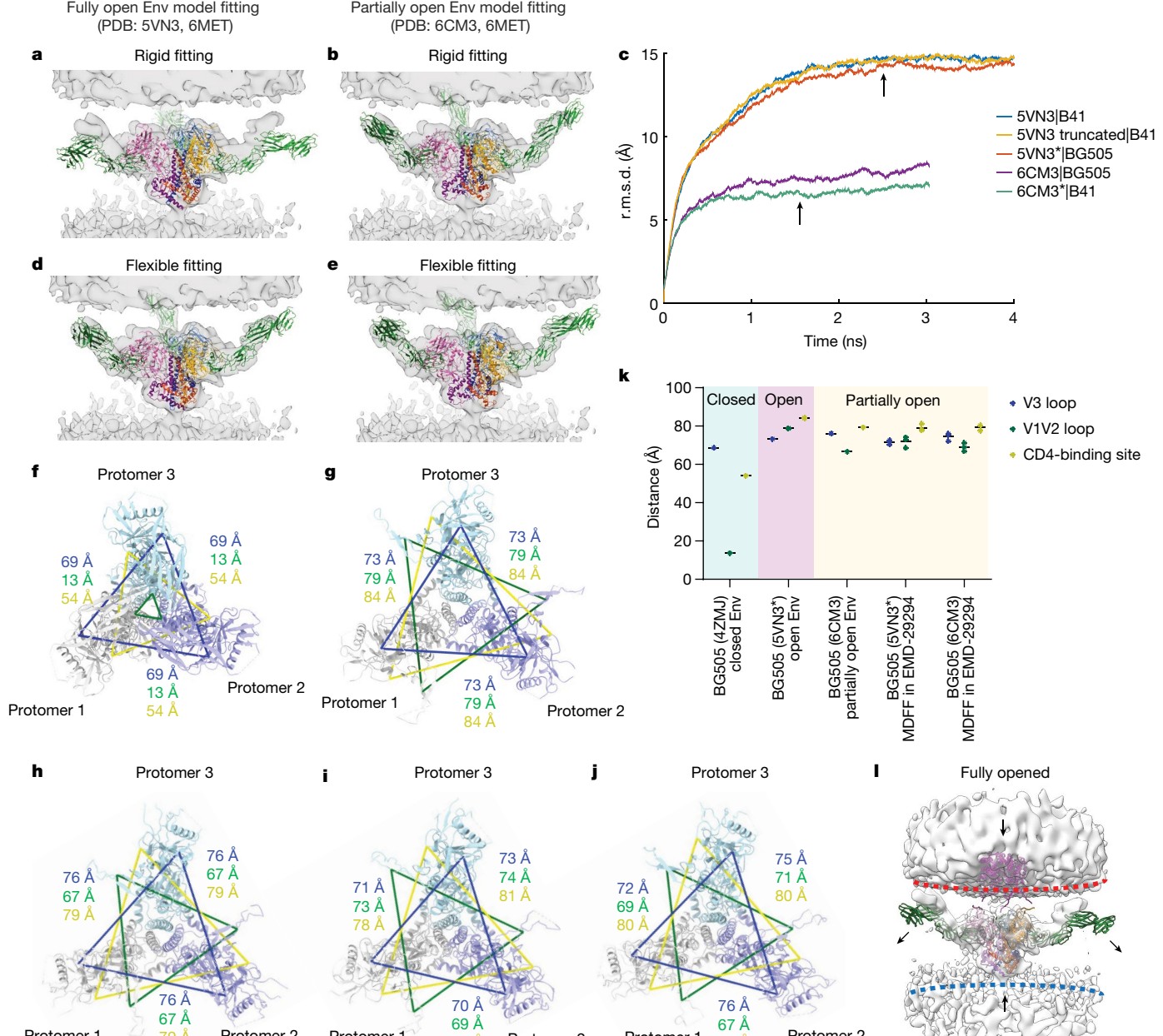

**Fig. 4 | MDFF analysis suggests that the Env trimer bound to three CD4 molecules is in a partially open conformation. a**,**b**, Rigid-body fitting of the SOSIP models (fully open Env (PDB: 5VN3; **a**)) and partially open Env (PDB: 6CM3; **b**)) containing full-length CD4 molecules (PDB: 6MET) into the cryo-ET density map of Env bound to three CD4 molecules. **c**, Time evolution of the backbone r.m.s.d. from five independent MDFF simulations into the same density, relative to their starting structure. MDFF simulations ran for around 4 ns on three models of open Env with CD4 (5VN3, truncated 5VN3 (SOSIP.651) and BG505 homology model 5VN3*) and for 3 ns on two models of partially open Env with CD4 (6CM3 and B41 homology model 6CM3*). The MDFF structures measured in **i** and **j** were selected after the MDFF simulation reached convergence (arrows; at 2.5 ns and 1.5 ns, respectively). **d**,**e**, The results of MDFF of the models in **a** (**d**) and **b** (**e**) into the same cryo-ET density map.

**f**–**k**, Analysis of Env openness on the models of BG505 SOSIP: closed (**f**, 4ZMJ), open (**g**, 5VN3*), partially open (**h**, 6CM3), and MDFF-fitted models (5VN3* fitted into Electron Microscopy Data Bank (EMDB) EMD-29294 (**i**) and 6CM3 fitted into EMD-29294 (**j**)). CD4 was omitted for clarity. Protomers are coloured in grey, purple and teal. The interprotomer distance was measured between α-carbons of three residues from each protomer: His330 (blue; located at base of the V3 loop), Pro124 (green; located at base of the V1V2 loop) and Asp368 (yellow; located at CD4-binding site). The distance and their average were plotted in **k**. **l**, Hypothetical conformational changes of the Env–CD4 complex required to engage the co-receptor. To overcome the steric constraints in CD4 molecules, release of CD4 or membrane bending may facilitate the movement of Env towards the coreceptors embedded in the target membranes.

(PDB: 6U0L) as fully open Env trimers[10,13] and BG505.SOSIP.664 (PDB: 6CM3) as a less-open Env trimer that was apparently partially closed due to binding of 8ANC195, a gp120–gp41 interface antibody[11]. To evaluate whether these structures fit into our cryo-ET density of the Env trimer bound to three CD4 molecules, full-length CD4 molecule (D1–D4), obtained from the model of the gp120 monomer bound to

CD4 molecule and coreceptor CCR5 (PDB: 6MET)[17], was superimposed onto the SOSIP models. Rigid-body fitting of the combined models into our density map indicated a better fit with the partially open Env model compared with a fully open Env model (Fig. 4a,b). We next performed molecular dynamics flexible fitting (MDFF)[35] analysis of the combined models within the cryo-ET density map. To facilitate a more

accurate comparison between the fitted models derived from B41. SOSIP.664 and BG505.SOSIP.664, three additional initial models were generated on the basis of the homology (*) and sequence length (truncated) (5VN3|truncated, 5VN3*|BG505 and 6CM3*|B41). MDFF analysis of modified initial models yielded similar outcomes to those obtained from the original models (Fig. 4c). Despite a residual root mean squared deviation (r.m.s.d.) difference of about 5 Å between the 6CM3 and 5VN3 models after MDFF (Extended Data Fig. 7a), probably attributed to resolution limitations in our cryo-ET map, both models demonstrated a good fit into the cryo-ET density map after MDFF (Fig. 4d,e, Extended Data Fig. 7b–h and Supplementary Videos 1 and 2). Generally, fully open models of Env derived from PDB 5VN3 had a larger r.m.s.d. shift (~15 Å) and required more time to converge (~2 ns), whereas partially open Env models derived from PDB 6CM3 showed a smaller r.m.s.d. shift (~7 Å) and achieved convergence in a shorter time frame (~1 ns) (Fig. 4c, Extended Data Fig. 7f,g and Extended Data Table 1). The shorter convergence time observed with the models derived from PDB 6CM3 suggests that the structure of Env bound to three CD4 molecules, as determined by cryo-ET, resembles a previously determined partially open CD4-bound soluble Env conformation.

To assess the openness of Env trimers in these models, we measured the inter-protomer distances between the Cα atoms of specific residues in the V1V2 loop (Pro124), V3 loop (His330) and CD4-binding site (Asp368). Consistent with model fitting, the models obtained through MDFF in the cryo-ET density map were consistent with the partially open Env conformation as seen in PDB 6CM3 (Fig. 4f–k). Together, these analyses indicate that partially open trimers are observable intermediates during HIV-1 Env binding to membrane-bound CD4. We postulate that partially open trimers are longer-lived intermediates when complexed with membrane-bound CD4 than with soluble CD4. This hypothesis is consistent with the fact that we did not observe shedding of gp120 into the supernatant when HIV-1$_{BaL}$ was incubated with MLV-CD4 (Extended Data Fig. 1) or any apparent loss of Env trimers at the membrane–membrane interfaces, whereas soluble CD4 induced shedding of gp120 (Extended Data Fig. 1) and many Env trimers were lost when HIV-1$_{BaL}$ was incubated with soluble CD4 and 17b[19]. These observations are also consistent with a previous study showing that soluble CD4 and CD4 mimetics trigger a short-lived activated intermediate of HIV-1 Env, whereas Env bound to cell-surface CD4 is long-lived[36].

## Discussion

Here we used cryo-ET to visualize the initial steps of HIV-1 entry whereby Env trimers engage CD4 receptor molecules residing in target membranes. Mixing HIV-1$_{BaL}$ virions with VLPs presenting CD4 receptors provided an experimental system that generated a high frequency of Env–CD4 complexes, enabling us to quantitatively study their interactions in biological membranes using cryo-ET. We observed HIV-1 Env trimers forming clusters and rings at membrane–membrane interfaces. As the distance between membranes at membrane–membrane interfaces decreased, structures of Env bound to increasing numbers of CD4 molecules became visible. This suggests that these images represent snapshots of a dynamic stepwise binding process.

The cluster and ring distributions of Env at membrane–membrane interfaces is highly similar to the organization of SNARE proteins at membrane-docking sites of synaptic vesicles[37]. It is possible that the patterns of Env–CD4 complexes observed at membrane–membrane interfaces may be a consequence of simple adhesion. However, both SNAREs and HIV-1 Env trimers are fusion machines that ultimately bring membranes together for fusion, suggesting that these observations are relevant for fusion.

The mixed viral particle system enabled us to visualize membrane–membrane interfaces containing Env–CD4 complexes at a high frequency. Acquiring 168 tomograms enabled us to identify 857 membrane–membrane interfaces containing around 5,700 Env–CD4

complexes for subtomogram averaging and subsequent classification. Our comprehensive analysis revealed that HIV-1 Env trimers engage a single CD4 when the two membranes are further apart and bind to a second and third CD4 as Env moves closer to the membrane. The conformational states of the Env trimers bound to one and two CD4 molecules are asymmetric, not just with respect to the inherent asymmetry in CD4 binding, but also with respect to the conformational state of each protomer within the trimer. The resolutions of our cryo-ET density maps were high enough to identify that the V1V2 loops project outward in all CD4-bound protomers, consistent with the protomer residing in the CD4-bound conformation. The density was less defined for the V1V2 loops in the unbound protomers of the Env trimers with one or two bound CD4 molecules indicating a distinct conformational state. Current limitations in the resolution that can be achieved by cryo-ET will not allow us to resolve the placement of the V1V2 loops at the apex, which are also known to be highly dynamic for tier 1b isolates[38]. The accompanying report[39] describes a model in which the V1V2 loops are localized at the trimer apex in these protomers. Cryogenic electron microscopy single-particle analysis (cryo-EM SPA) was used to solve high-resolution structures of soluble BG505 SOSIP trimers bound to one and two CD4 molecules[39]. In the model with two CD4 molecules bound to Env (BG505 HT2), the CD4-bound protomers adopted the typical CD4-bound conformation with V1V2 loops displaced from the Env apex to the sides of gp120 with a fully formed four-stranded antiparallel bridging sheet. By contrast, the unbound gp120 moved by full-body rotation but the V1V2 loops remained at the trimer apex, adopting an occluded open conformation[39]. The soluble BG505 HT2 model bound to two CD4 molecules[39] fit well into the densities that we observed using cryo-ET. The predominant structure of the BG505 HT1 trimer bound to one CD4 molecule was a closed trimer in ref. 39, but an open trimer in our biological membranes. However, subclassification in the accompanying report[39] revealed that about one-third of the BG505 HT1 trimers bound to one CD4 adopted an open trimer conformation that superimposed well onto our cryo-ET structure of Env bound to one CD4 molecule[39]. The results from both laboratories are therefore consistent, particularly as it is known that a more CD4-resistant tier 2 HIV-1 isolate such as BG505 is less responsive to CD4 than the tier 1b isolate HIV-1$_{BaL}$ used for cryo-ET[10,38]. The similar structures determined by cryo-ET and cryo-EM SPA demonstrate the existence of asymmetric intermediates of Env during CD4 receptor binding.

Previous smFRET studies performed by our group predicted the existence of asymmetry in the opening of the trimer[22]. Trimers engineered to bind to only one CD4 molecule featured the neighbouring unliganded protomers in an intermediate conformational state that was distinct from the unbound and CD4-bound conformations[22]. Moreover, smFRET suggested that HIV-1 Env has spontaneous access to this intermediate conformational state and its occupancy increased in response to CD4 binding[22]. These agreements raise the hope that parallel cryo-EM and smFRET studies can resolve remaining unknown questions about the states of Env observed by both methods. One such unresolved question relates to an additional conformational state of HIV-1 Env on the surface of virions predicted by smFRET[21,40]. This state is fragile and affected by mild treatments including AT-2 inactivation[19]. As AT-2-treated virus particles are on path along Env conformational transition during fusion, show no defect in their ability to bind to CD4 receptor and fuse with cells[20,26], this unknown conformational state is probably upstream of interactions with CD4 studied in this report.

The detection of asymmetric HIV-1 Env trimers bound to one and two CD4 molecules demonstrate that these intermediates are quite common and long lived, which probably has consequences for antibody-mediated immune responses and vaccine immunogen design. The present study and the accompanying report[39] reveal a mechanism of trimer opening that generally maintains immune evasion principles by keeping the V1V2 loops on top of the apex hindering access to

epitopes for non-neutralizing antibodies. However, antibodies that bind to these partially open conformational states have recently been identified, which suggests that the V1/V2 epitopes on these intermediates are potentially recognized by the immune system[41,42].

Our studies in this and the accompanying report[39] point to a stepwise model of HIV-1 binding to the CD4 receptor (Fig. 4l). Initial contact between HIV-1 virions and cells is mediated by binding of a single protomer to an elongated CD4 receptor molecule when the membranes are still around 20 nm apart from each other. Engagement of a second and third CD4 molecule enables HIV-1 Env to move closer to the target membrane. This movement is facilitated by conformational changes within CD4 whereby the D3D4 domains align with the target membrane. However, even with these conformational changes, the Env trimer bound to three CD4 molecules remains too far away to engage the membrane-embedded co-receptor (Fig. 4l). There are considerable steric constraints in the gp120–CD4 interaction that must be overcome for the Env trimer to bind to the co-receptor. This could be achieved by either release of CD4 as gp120 transitions to the coreceptor or membrane bending in the target cell. Future experiments with CCR5 embedded in target membranes will shed light on co-receptor binding and subsequent formation of the prehairpin intermediate. The experimental system described here may assist in characterizing these subsequent steps of the HIV-1-entry process.

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

## Methods

### Plasmids, cell lines and other reagents

HIV-1$_{BaL}$/SUP-T1-CCR5 cells (CLN204)[43] were cultured in RPMI-1640 medium supplemented with 10% fetal bovine serum (FBS), 100 U ml$^{-1}$ penicillin, 100 µg ml$^{-1}$ streptomycin and 2 mM L-glutamine in the presence of 5% CO$_2$. HEK293T, ADA.CM.755*/HEK293T and TZM-bl cells were cultured in DMEM medium supplemented with 10% FBS, 100 U ml$^{-1}$ penicillin, 100 µg ml$^{-1}$ streptomycin and 2 mM L-glutamine in the presence of 5% CO$_2$. Cells were transfected at 60–80% confluency and the culture medium was exchanged before transfection. HIV-1 *gag-pol* was expressed by pCMV ΔR8.2 (Addgene plasmid, 12263). The protease mutations D25N and R57G were generated by overlapping PCR using pCMV ΔR8.2 as a template. The mutations were validated by Sanger sequencing. Human CD4-expressing vector pcDNA-hCD4 was provided by H. Gottlinger. The MLV GagPol plasmid was a gift from R. Mulligan. The following reagents were obtained through the NIH HIV Reagent Program, Division of AIDS, NIAID, NIH: indinavir sulfate (ARP-8145) and ritonavir (ARP-4622), contributed by DAIDS/NIAID.

### Gp120-shedding assay

HIV-1$_{BaL}$ viruses were incubated at room temperature for 1 h with PBS, soluble CD4 (100 µg ml$^{-1}$) alone, soluble CD4 (100 µg ml$^{-1}$) in combination with 17b Fab (100 µg ml$^{-1}$), or MLV-CD4 VLPs. After incubation, the samples were ultracentrifuged at 130,000*g* through a 20% sucrose cushion for 1 h to separate the supernatant and pellet fractions. Western blotting was performed using the 2G12 antibody (ARP-1476, NIH HIV Reagent Program) to detect gp120 and anti-HIV-1 serum (ARP-1983, NIH HIV Reagent Program) to detect gp41 and p24, HRP goat anti-human Ig (2010-05, SouthernBiotech) was used as the secondary antibody.

### Sample preparations for cryo-ET

HIV-1$_{BaL}$ virus (P4311) was produced using the HIV-1$_{BaL}$/SUP-T1-CCR5 cell line and prepared as previously described[19]. In brief, 30 l of cell culture was grown in roller bottles (1 rpm) incubated at 37 °C in the absence of CO$_2$ and the product was serially filtered using 5 µm capsule filters to remove the cells, followed by filtration using 0.45 µm capsule filters to remove cell debris and large microvesicles. The filtrate was treated with a final concentration of 1 mM 2,2′-dithiodipyridine (AT-2). AT-2 eliminates retroviral infectivity by mediating the covalent modification of free cysteines on internal viral proteins, including the zinc-finger cysteine residues on the viral nucleocapsid protein. Even after AT-2 treatment, the envelope glycoproteins with disulfide-bonded cysteines on the virion surface retain the ability to bind to CD4, undergo conformational changes and mediate membrane fusion[20,26]. HIV-1$_{BaL}$ viruses were next purified by continuous-flow sucrose-density-gradient ultracentrifugation.

Immature HIV-1$_{BaL}$ viruses were produced by the HIV-1$_{BaL}$/SUP-T1-CCR5 cell line treated with a final concentration of 1 µM indinavir and 10 µM ritonavir. Cells were treated for 2 days, after which the cell medium was discarded and replaced with fresh medium with the same concentrations of indinavir and ritonavir for two additional days. The cell culture medium was then collected and clarified by low-speed centrifugation at 1,500 rcf for 15 min, followed by filtration with 0.45 µm syringe filters (Pall Corporation). The filtered supernatants were loaded into 38.5 ml open-top ultracentrifuge tubes (Beckman Coulter) and underlayed with 5 ml of 15% sucrose in PBS. The viruses were then pelleted by ultracentrifugation at a maximum of 131,453 rcf using a Beckman SW28 swinging bucket rotor at 27,000 rpm for 1 h at 4 °C. The supernatant was removed and viruses were resuspended in 1/1,000 volume of PBS.

HIV-1$_{ADA.CM.755*}$ was produced from the ADA.CM.755*/HEK293T cell line[33] by transfection with 10 µg DNA per 10 cm plate of the HIV-1 GagPol plasmid pCMV ΔR8.2 using polycation polyethylenimine (PEI) (pH 7.0, 1 mg ml$^{-1}$). The cell culture medium was collected 2 days after transfection, then clarified and filtered as described above. Viruses were pelleted by ultracentrifugation and resuspended in PBS as described above.

Immature HIV-1$_{ADA.CM.755*}$ viruses were produced from the ADA.CM.755*/HEK293T cell line by transfecting pCMV ΔR8.2 containing the protease mutations D25N or R57G. Viruses were collected after 2 days using the same methods as described above.

MLV particles carrying CD4 receptor (MLV-CD4 VLPs) were produced by co-transfecting plasmids encoding MLV GagPol and CD4 at a 1:1 ratio into HEK293T cells using PEI. Viruses were collected after 2 days using the same methods as described above.

Plasma membrane blebs were produced from the TZM-bl cell line according to a previously described protocol[44]. In brief, when cells reached around 70% confluency, the medium was removed and the cells were washed twice with GPMV buffer (10 mM HEPES, 150 mM NaCl, 2 mM CaCl$_2$, pH 7.4). Subsequently, the cells were incubated with blebbing buffer (GPMV buffer containing 25 mM PFA and 2 mM DTT) at 37 °C overnight. The blebs were then collected from the supernatant the next day followed by a short centrifugation (1,500*g* for 5 min) to remove the cell debris. The blebs were concentrated using Amicon Ultra 15 ml centrifugal Filters (UFC910024, Millipore) to ~500 µl, and then extruded through 200 nm filters using the Mini Extruder (Avanti Polar Lipids) according to the manufacturer's protocol. The extruded blebs were further concentrated using Amicon Ultra 0.5 ml centrifugal Filters (UFC510096, Millipore) to about 20 µl. The freshly prepared concentrated blebs were used for cryo-ET sample preparation.

### Cryo-ET sample preparation

HIV-1$_{BaL}$ particles and blebs or MLV-CD4 VLPs were mixed and incubated at room temperature for 30 min. Next, a 6 nm gold tracer was added at a 1:3 ratio. A total of 5 µl of the mixture was incubated for 1 min on freshly glow-discharged holey carbon grids (Quantifoil R 2/1 on 200 copper mesh). The grids were blotted with filter paper and plunge-frozen into liquid ethane using a homemade gravity-driven plunger apparatus. Frozen grids were stored in liquid nitrogen before imaging.

### Cryo-ET data collection

Cryo-grids were imaged on the Titan Krios G2 Cryo-transmission electron microscope (Thermo Fisher Scientific) operated at 300 kV using either a K2 or K3 direct electron detector (Gatan) in counting mode with a 20 eV energy slit and a Volta phase plate[45]. Tomographic tilt series between −51° and +51° were collected using SerialEM[46] with a 3° step size and a dose-symmetric scheme[47,48]. The nominal magnification for the K2 direct electron detector was ×105,000, giving a pixel size of 1.333 Å, while, for the K3 direct electron detector, it was ×64,000, giving a pixel size of 1.346 Å. The raw images were collected from single-axis tilt series with a cumulative dose of 123 e$^-$ Å$^{-2}$. The defocus was set at −0.5 µm. Frames were motion-corrected using Motioncorr2 (ref. 49) to generate drift-corrected stack files, which were then aligned using gold fiducial makers using Etomo[48]. Weighted back projection and tomographic slices were visualized using IMOD[48].

### Cryo-ET data analysis

To analyse the membrane–membrane interfaces, two coordinate points were placed at the point of closest approach between opposing membranes. These two coordinate points were used to quantify the distances between the membranes. Subtomograms of the interfaces were extracted and centred on the middle point of two membrane coordinate points. Averaging processing was performed with I3[50] with 8× binned tomograms. Env–CD4 complexes at the membrane–membrane interfaces were picked manually in IMOD. Initial Euler angles were determined on the basis of the vector between two coordinate points of the opposing membranes. Subtomograms were extracted for initial alignment. Subsequent processing was performed using I3 with 4× and 2× binned tomograms. The coordinates of Env–CD4 complexes

were overlaid onto the averaged interfaces on the basis of the Euler angles obtained after alignment. Custom Python scripts were used to generate kernel density heat maps and histogram profiles fit with Gaussian curves depicting the distributions of Env–CD4 complexes. All of the density maps were visualized and segmented in the UCSF ChimeraX[51]. The Fourier shell correlation curves were calculated using Relion[52] and the local resolutions of averaged structures were determined using ResMap[53].

To quantify Env clustering, all Env trimers on virions were manually picked. The arc distance between any two Env trimers on each individual virion was calculated using a custom-made script based on the approximate centre of the virion and the Env positions after initial subtomogram averaging. The formula used for calculating the arc distance was $s = r\theta$, where $s$ represents the arc distance between two Env trimers, $r$ is the radius of the virion and $\theta$ is the angle subtended by the arc at the centre of the virion. The histogram of Env distances was plotted in GraphPad Prism v.9.0. A custom script was used to conduct multiple distance spatial cluster analysis using the linear distances between Env trimers. The script calculated and plotted the ratio of the largest number of Env trimers within increasing cluster radii to the total Env number on individual virions.

### MDFF

The cryo-EM density map obtained from subtomogram averaging was low-pass filtered to 15 Å resolution. The fully open model of CD4-bound B41 SOSIP trimer (from PDB 5VN3) and the partially open model of CD4-bound BG505 SOSIP trimer (from PDB 6CM3) were first fit into the density map by rigid-body fitting in Chimera[54]. The full-length CD4 molecules from the CD4–gp120–CCR5 complex (PDB: 6MET) were then added to both SOSIP trimers 5VN3 and 6CM3 by aligning and superimposing the gp120 subunits to build the initial models of Env–CD4 complex. The coordinates of the initial models were saved relative to the density map. To ensure the comparability of the fitting result, we incorporated three additional models, which are a homology model based on BG505.SOSIP sequence using PDB 5VN3 as the template (referred to as 5VN3*|BG505), a homology model based on the B41.SOSIP sequence using PDB 6CM3 as the template (referred to as 6CM3*|B41) and a truncated version of the PDB 5VN3 model (B41. SOSIP.651) (referred to as 5VN3 truncated|B41). The homology modelling was performed using SWISS-MODEL in user template mode[55]. Initially, the full-length sequence was used, followed by removal of the residues that were missing in the templates, and editing of the chain name and residue numbers. The CD4 molecules were added using the same approach as in the original models. Consequently, 5VN3* shares the exact atom and chain name with the 6CM3 model. However, the 6CM3* model has shorter gp120 and gp41 chain compared with 5VN3, as the template 6CM3 consists of corresponding shorter chains. As a result, the truncated 5VN3 model was constructed specifically for comparison to 6CM3*.

The initial models for MDFF were then prepared using the Visual Molecular Dynamics program[56] to incorporate additional elastic restraints for secondary structure dihedrals, hydrogen bonds, chiral centres and *cis*-peptide bonds. Those restraints ensured the maintenance of proper secondary structure and prevented overfitting.

The MDFF simulations were conducted using NAMD v.3.0 alpha[57] with CHARMM36 force-field parameters[58], at a temperature of 300 K in vacuo. The partially open model underwent MDFF for a duration of 3 ns with a scaling factor ($\xi$) of 0.1, followed by an energy minimization step using a scaling factor $\xi = 0.5$ over a time span of 0.05 ps. Similarly, the fully open model was processed for MDFF for 4 ns with $\xi = 0.1$ followed by energy minimization over 0.05 ps using a scaling factor $\xi = 0.5$.

### Statistical analysis

Data were analysed and plotted using GraphPad Prism. Statistical significance for pairwise comparisons was derived using two-tailed nonparametric Mann–Whitney tests. To assess statistical significance for multiple comparisons we used Kruskal–Wallis tests followed by Dunn's multiple-comparison test. $P < 0.05$ was considered to be statistically significant; $*P < 0.05$, $**P < 0.01$, $***P < 0.001$, $****P < 0.0001$.

### Reporting summary

Further information on research design is available in the Nature Portfolio Reporting Summary linked to this article.

## Data availability

The cryo-EM structures have been deposited at the EMDB. Cryo-ET structural maps for HIV-1 Env bound to one, two and three CD4 molecules, the consensus structure of HIV-1 Env bound to CD4 receptors in membranes and the unliganded Env trimer have been deposited under accession codes EMD-29292, EMD-29293 and EMD-29294, EMD-29295 and EMD-41045, respectively.

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

**Acknowledgements** We thank K. Dam, C. Fan and P. Bjorkman for sharing unpublished single-particle cryo-EM structures and reading the manuscript; S. Wu for her technical assistance; and J. Binley, H. Gottlinger, R. Mulligan, J. Sodroski and Z. Matsuda for plasmids. This work was supported by NIH grants R37 AI150560 and U54 AI170856 to W.M., R01 AI143563 to M.B.Z. and R01 GM125769 to H.D.T. E.N. and M.W.G. are supported by NIH T32AI055403. M.W.G. is a recipient of the Gruber Science Fellowship. J.D.L. and J.W.B.J. are supported in part by federal funds from the National Cancer Institute, National Institutes of Health, under contract no. 75N91019D00024/HHSN261201500003I.

**Author contributions** W.L., E.N. and W.M. conceived experiments. W.L., E.N., J.W.B.J., J.D.L., J.R.G., M.B.Z. and P.D.U. produced HIV-1 and MLV viral particles. W.L. and M.W.G. collected cryo-ET data. W.L., Z.Q., M.W.G., E.N. and H.D.T. analysed cryo-ET data. Z.Q. performed the MDFF. E.N., W.L. and W.M. wrote the original manuscript draft. All of the authors reviewed and edited manuscript. W.L. and W.M. supervised the work. W.M. acquired funding.

**Competing interests** The authors declare no competing interests.

**Additional information**
**Correspondence and requests for materials** should be addressed to Wenwei Li or Walther Mothes.

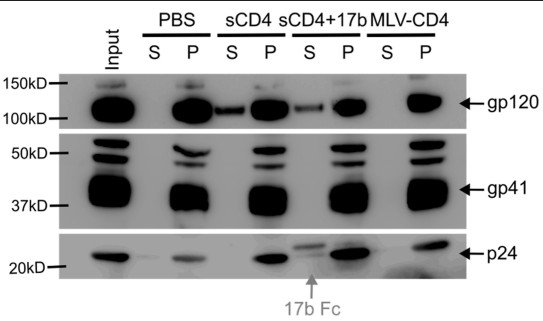

**Extended Data Fig. 1 | sCD4 and sCD4 + 17b induce gp120 shedding on HIV-1$_{BaL}$ virions.** Western blotting analysis for gp120, gp41 and p24 in the supernatant (S) and pellet (P) fractions of HIV-1$_{BaL}$ viruses from ultracentrifugation after incubation with PBS, sCD4 alone, sCD4 in combination with 17b Fab, or MLV-CD4 VLPs. Residual 17b Fc fragments in 17b Fab were detected by anti-Human 2nd HRP-antibody (grey).

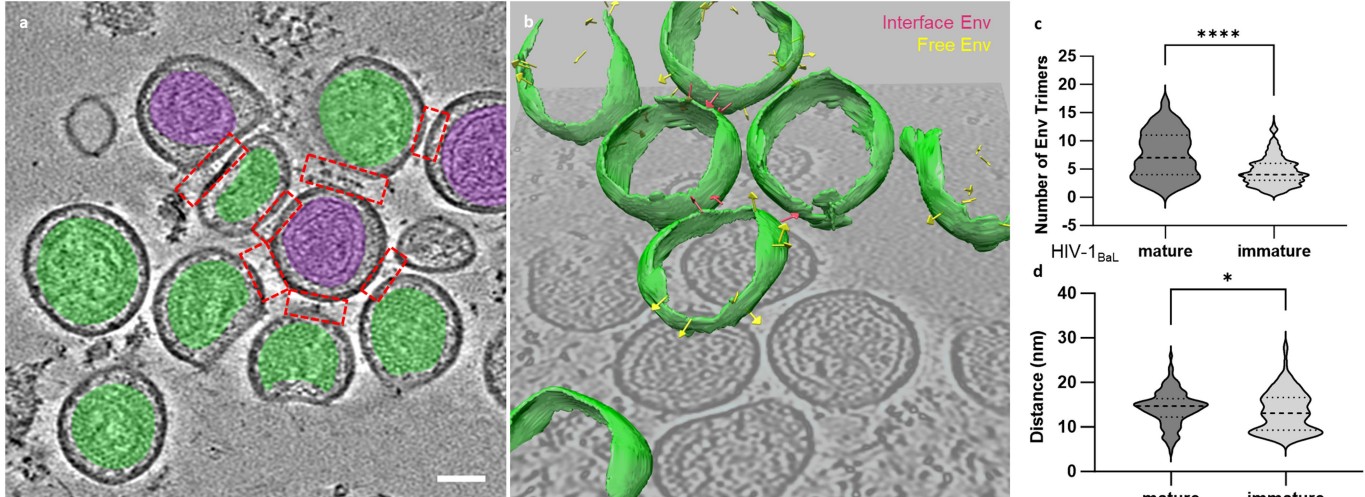

**Extended Data Fig. 2 | Env-CD4 interactions at the membrane-membrane interfaces between immature HIV-1$_{BaL}$ particles and MLV-CD4 VLPs.**
**a** A representative tomogram showing immature HIV-1$_{BaL}$ particles and MLV-CD4 VLPs. HIV-1 capsids are noted in green and MLV capsids are noted in purple. Scale bar = 50 nm. **b** Three-dimensional representation of immature HIV-1BaL particles and MLV-CD4 VLPs in a representative tomogram. Pink arrows represent Env trimers at membrane-membrane interfaces. Yellow arrows represent free Env trimers on the surface of the HIV-1BaL particles. **c** The number of Env trimers present at membrane-membrane interfaces for mature and immature HIV-1$_{BaL}$ particles mixed with MLV-CD4 VLPs. Mature HIV-1$_{BaL}$ mean ± s.d. = 7.5 ± 4. Immature HIV-1$_{BaL}$ mean ± s.d. = 4.6 ± 2.5. P-value **** = <0.0001. **d** The distance between membranes at membrane-membrane interfaces with mature and immature HIV-1$_{BaL}$ particles mixed with MLV-CD4 VLPs. Mature HIV-1$_{BaL}$ mean ± s.d. = 14.3 nm ± 3.8 nm. Immature HIV-1$_{BaL}$ mean ± s.d. = 13.5 nm ± 4.6 nm. * = 0.0420.

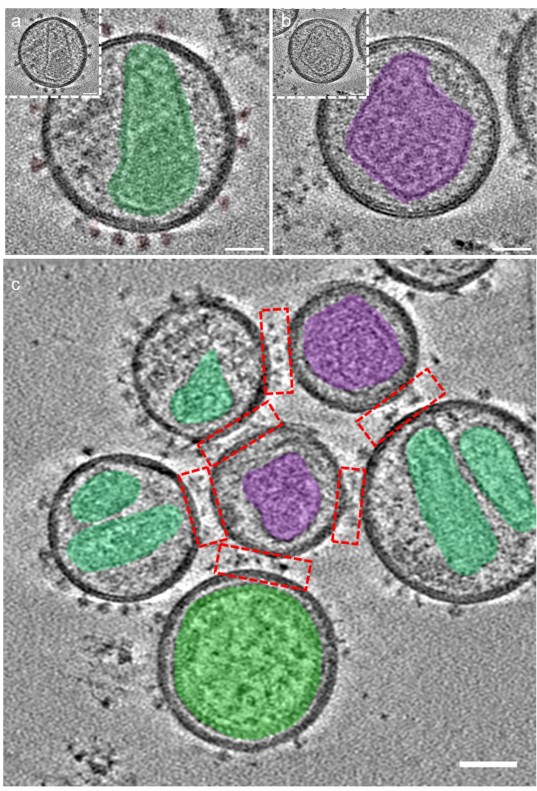

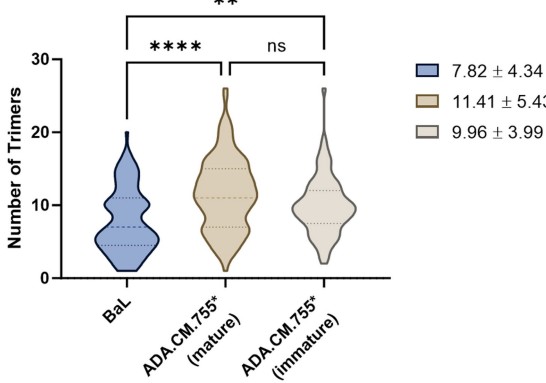

**d**

**Number of Trimers at Membrane-Membrane Interfaces**

- 7.82 ± 4.34
- 11.41 ± 5.43
- 9.96 ± 3.99

**Extended Data Fig. 3 | HIV-1 Env clustering is unaffected by capsid maturation in HIV-1 particles featuring a high number of Env trimers lacking the C-tail. a** Representative images of the HIV-1 viral particle produced from the cell line expressing $Env_{ADA.CM.755*}$. The capsid is highlighted in green. Scale bar = 25 nm. **b** Representative images of the MLV VLP carrying CD4. The capsid is highlighted in purple. Scale bar = 25 nm. **c** A representative image of membrane-membrane interfaces in cryo-tomograms. HIV-1 and MLV capsids are highlighted in green and purple, respectively. Scale bar = 50 nm. **d** Violin plot of the number of Env trimers present at membrane-membrane interfaces for HIV-1$_{BaL}$ (blue), mature HIV-1$_{ADA.CM.755*}$ (brown), and immature HIV-1$_{ADA.CM.755*}$ (beige) with MLV-CD4 VLPs. P-values ns = 0.0779, ** = 0.0028, **** = <0.0001.

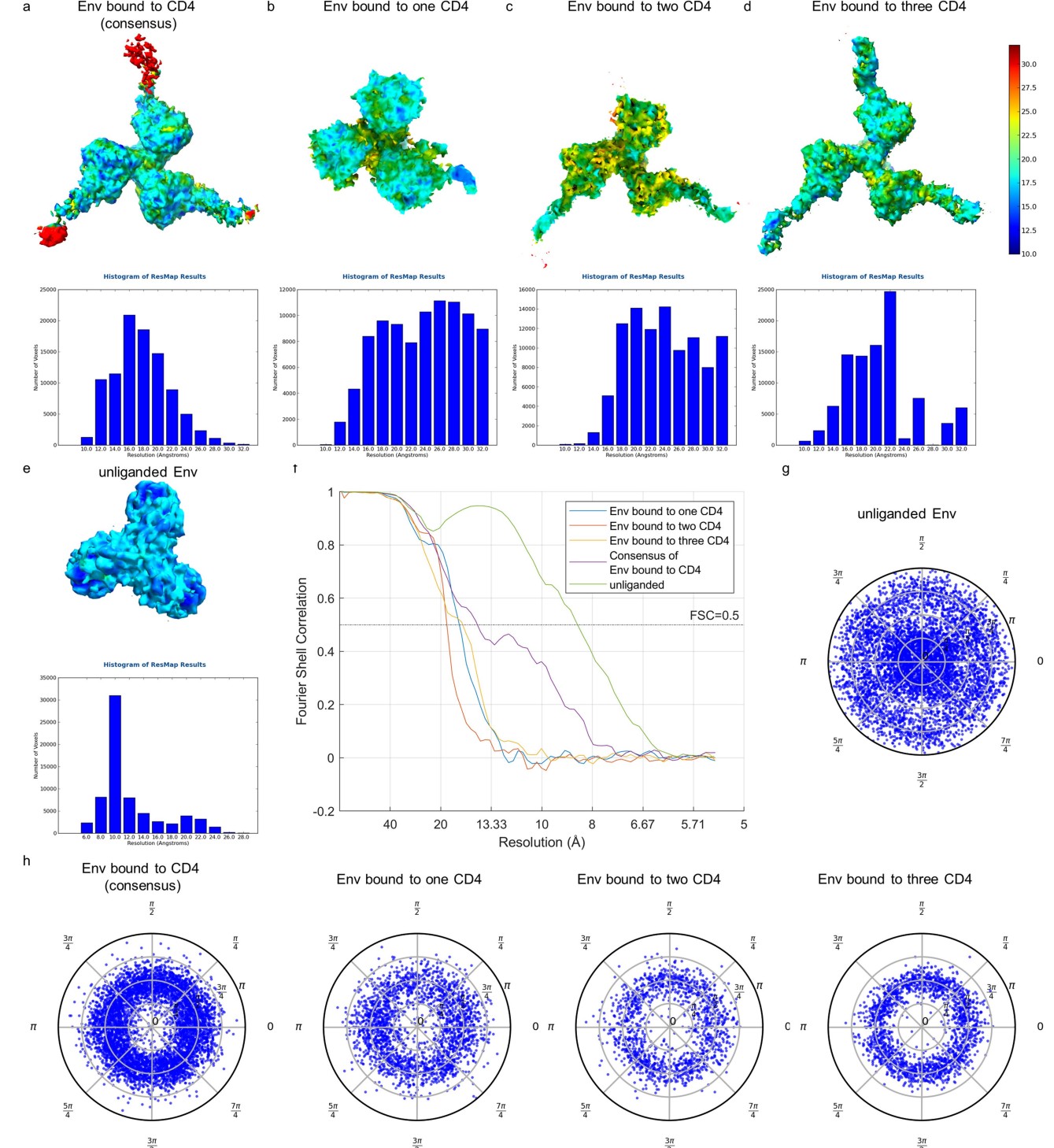

**Extended Data Fig. 4 | Resolution assessment of subtomogram averaged structures. a–e** Local resolution estimation using ResMap software for the structures of the Env-CD4 complex. The structures and ResMap histograms of the consensus average (**a**), Env trimer bound to one CD4 molecule (**b**), Env trimer bound to two CD4 molecules (**c**), Env trimer bound to three CD4 molecules (**d**) and unliganded Env trimer from the same dataset (**e**) are shown. Subtomogram averages are coloured according to the local resolution as indicated on the scale on the right. **f** Fourier shell correlation (FSC) curves for cryo-ET masked subtomogram averages with FSC = 0.5 as the cutoff value. **g–h**. The distribution of orientations for aligned Env particles. The unliganded Env particles are evenly distributed across all orientations (**g**). The Env particles bound to CD4 molecules showed reduced numbers of top-view and bottom-view particles (**h**). This is likely because the Env-CD4 complexes are located at the interface between two viruses, resulting in a higher occurrence of side-view orientations.

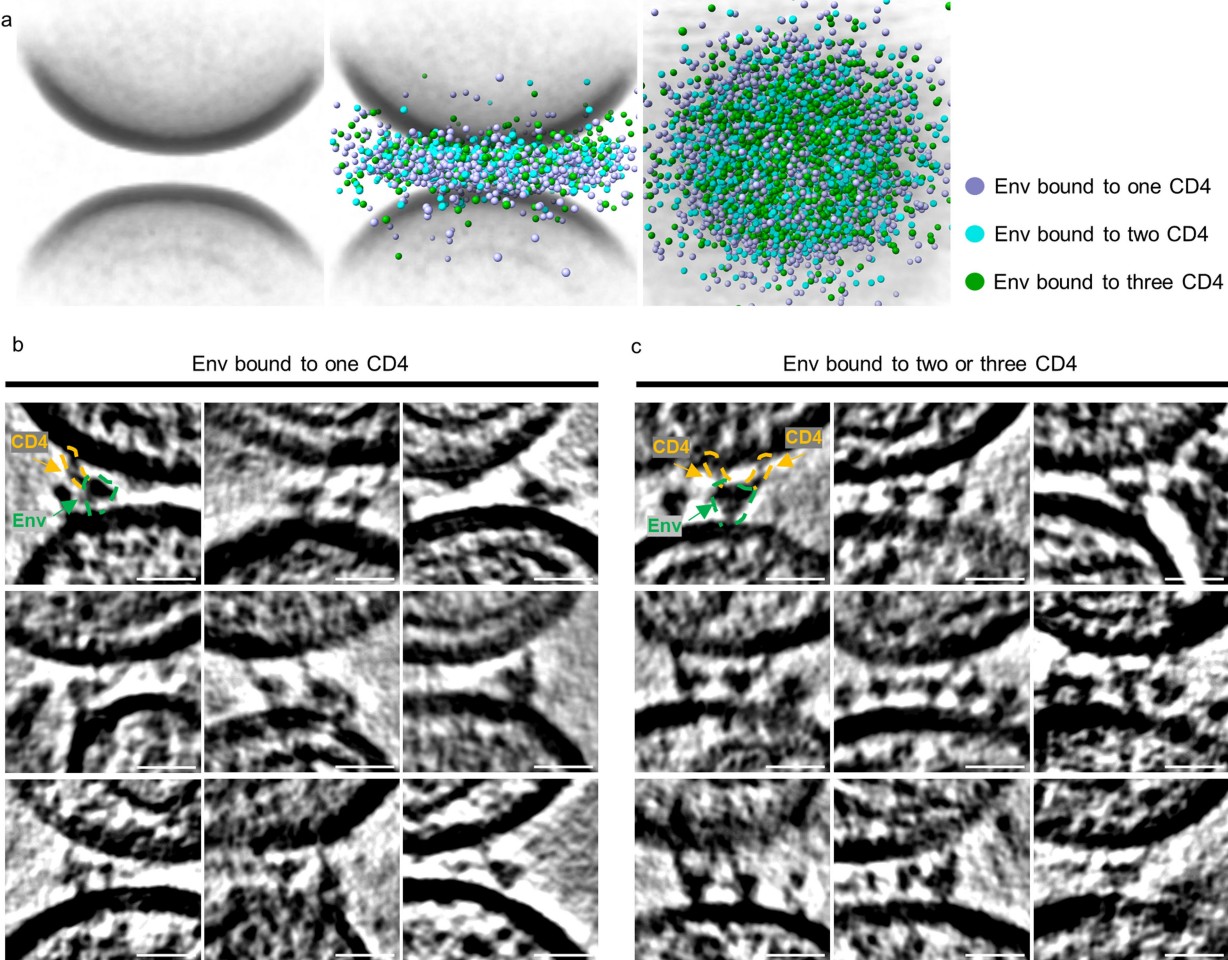

**Extended Data Fig. 5 | HIV-1 Env trimers bound to one, two and three CD4 receptors on biological membranes. a** Individual subtomograms of all the membrane-membrane interfaces were aligned and averaged (left panel). In the middle panel, the coordinates of Env bound to one (purple), two (cyan) and three (green) CD4 molecules are overlaid onto the averaged volumes.

The top-down view is shown to the right. **b, c** Gallery of HIV-1 Env trimers bound to CD4 receptors on biological membranes. Representative images from cryo-tomograms of the HIV-1 Env trimers bound to one CD4 (b) or two and three CD4 (c) molecules at the membrane-membrane interfaces between HIV-1 and MLV-CD4 particles. Scale bar = 20 nm.

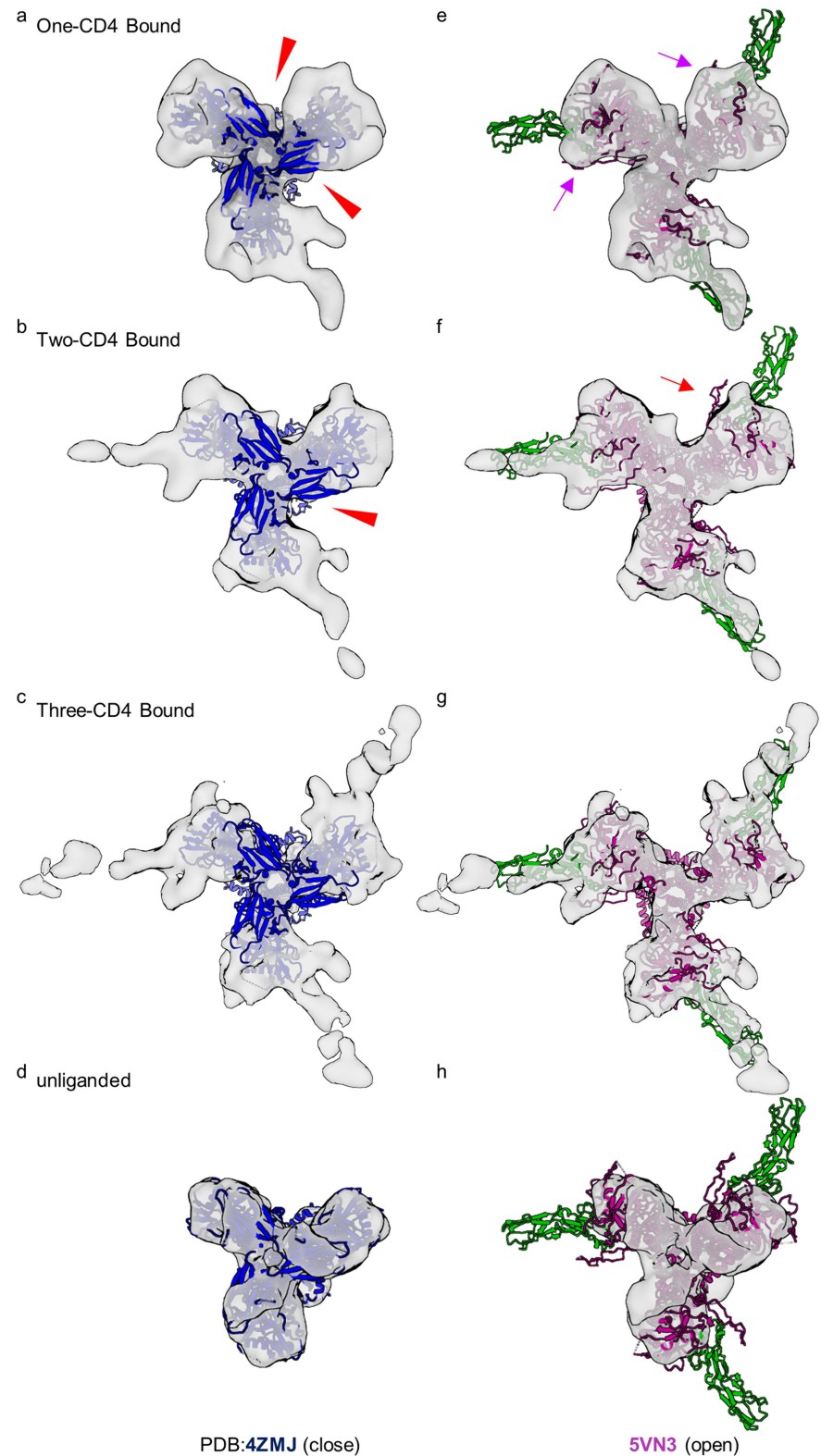

a One-CD4 Bound

b Two-CD4 Bound

c Three-CD4 Bound

d unliganded

e

f

g

h

PDB:**4ZMJ** (close)

**5VN3** (open)

**Extended Data Fig. 6 | HIV-1 Env trimers bound to one and two CD4 molecules are asymmetric with protomers adopting distinct conformational states.** **a**–**h** Rigid fitting of model of Env in a closed conformation (PDB:4ZMJ[59]) (**a**–**d**) and an open conformation (PDB:5VN3) (**e**–**h**) into the density maps of Env bound to one CD4 molecule (**a**, **e**), two CD4 molecules (**b**, **f**), three CD4 molecules (**c**, **g**) and unliganded Env (**d**, **h**). Missing densities at the apex region of non-CD4 bound protomers are indicated by red arrowheads (**a**, **b**). Weak densities of the V1V2 loops on the non-CD4 bound protomers in the Env trimer bound to one CD4 molecule are indicated by purple arrows (**e**). Lack of density for the outward projecting V1V2 loops on the non-CD4 bound protomer in the Env trimer bound to two CD4 molecules is indicated by a red arrow (**f**).

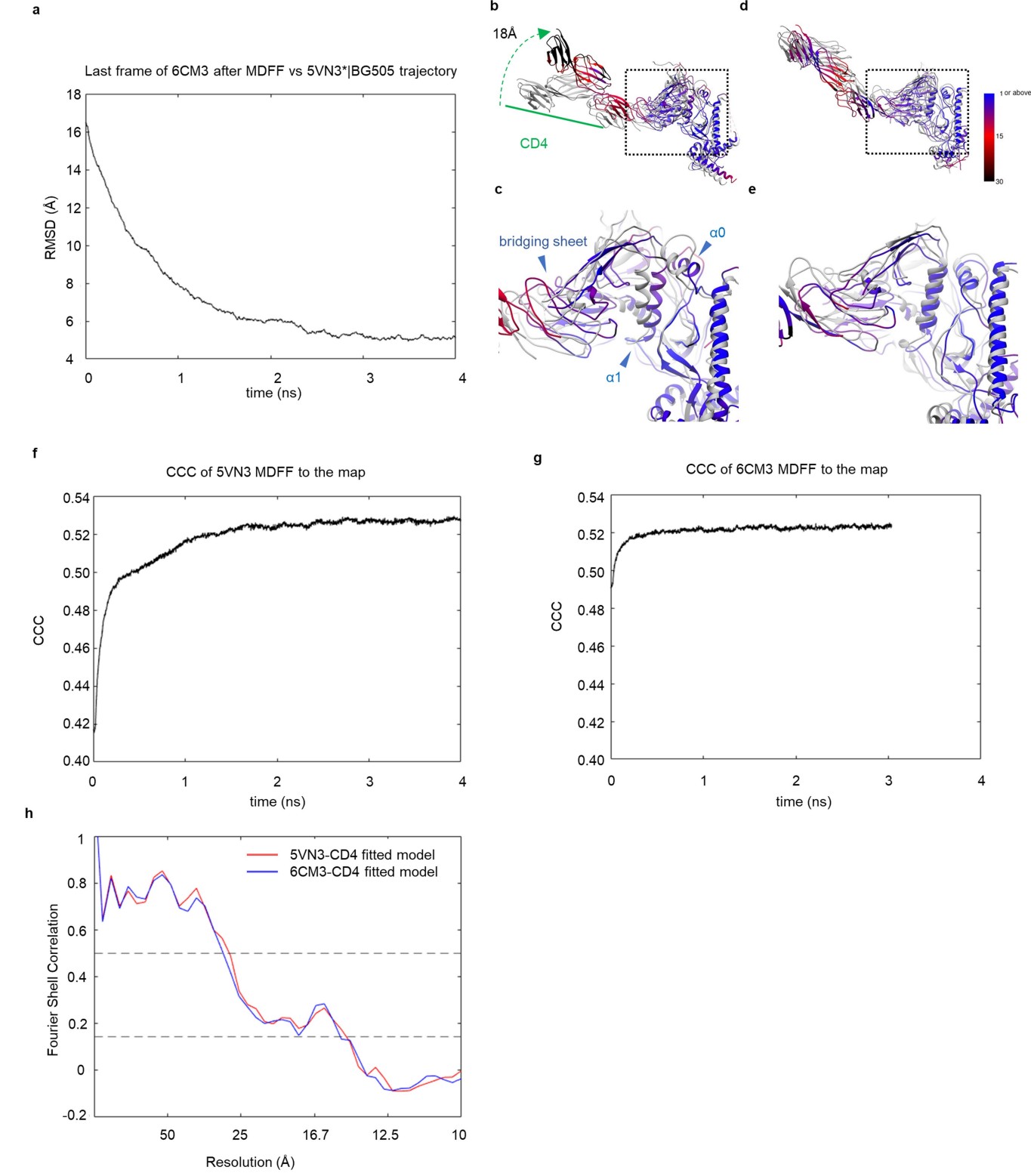

**Extended Data Fig. 7 | Molecular dynamics flexible fitting (MDFF) suggests that the Env trimer bound to three CD4 molecules is in a partially open conformation. a** The time evolution of backbone-RMSD of 5VN3* comparing to the last frame of MDFF simulation with 6CM3. **b**–**e** Atomic models showing the positions of CD4 (**b**, **d**) and gp120 (**c**, **e**) before (grey) and after (coloured) MDFF simulation with 5VN3 (**b**, **c**) and 6CM3 (**d**, **e**). The colour gradient bar indicates the amount of movement between the two structures. **f**, **g** The cross-correlation coefficient (CCC) curve between the density map and the trajectories of MDFF simulations (5VN3 in **f** and 6CM3 in **g**). **h** The map-model Fourier shell correlation (FSC) curve between the density map and the trajectories of MDFF simulations (5VN3 in red and 6CM3 in blue). The membranes in the maps were masked out for the analyses.

**Extended Data Table 1 | Shifts in the centre of mass for the domains of each Env protomer and CD4 molecule after flexible fitting (Å)**

| Domains<br>5VN3<br>6CM3 | Protomer1 | Protomer2 | Protomer3 |
|---|---|---|---|
| gp120 | 2.36 | 3.11 | 2.39 |
| | 1.16 | 2.89 | 1.77 |
| gp41 | 1.57 | 2.37 | 3.08 |
| | 2.13 | 3.36 | 1.84 |
| CD4 | 18.74 | 16.56 | 16.49 |
| | 7.46 | 8.31 | 6.75 |
| gp120 α0 Helix | 3.08 | 4.89 | 3.70 |
| | 2.10 | 1.39 | 2.77 |
| gp120 α1 Helix | 3.82 | 4.75 | 3.51 |
| | 1.63 | 2.97 | 1.95 |
| bridging sheet | 5.53 | 7.06 | 7.60 |
| | 1.47 | 4.59 | 3.96 |
| CD4 D1 | 7.12 | 5.44 | 5.90 |
| | 4.64 | 5.71 | 3.92 |
| CD4 D2 | 14.93 | 12.09 | 11.72 |
| | 9.29 | 11.52 | 9.00 |
| CD4 D3 | 22.72 | 19.94 | 20.57 |
| | 10.86 | 12.92 | 10.58 |
| CD4 D4 | 34.86 | 36.09 | 32.42 |
| | 9.00 | 8.39 | 8.77 |

Table of the shifts in the centre of mass for the domains of Env protomers and CD4 after the MDFF analysis in Fig. 4 and Extended Data Fig. 7.

# Reporting Summary

## Statistics

For all statistical analyses, confirm that the following items are present in the figure legend, table legend, main text, or Methods section.

| n/a | Confirmed | |
|---|---|---|
| ☐ | ☒ | The exact sample size (*n*) for each experimental group/condition, given as a discrete number and unit of measurement |
| ☐ | ☒ | A statement on whether measurements were taken from distinct samples or whether the same sample was measured repeatedly |
| ☐ | ☒ | The statistical test(s) used AND whether they are one- or two-sided *Only common tests should be described solely by name; describe more complex techniques in the Methods section.* |
| ☒ | ☐ | A description of all covariates tested |
| ☒ | ☐ | A description of any assumptions or corrections, such as tests of normality and adjustment for multiple comparisons |
| ☐ | ☒ | A full description of the statistical parameters including central tendency (e.g. means) or other basic estimates (e.g. regression coefficient) AND variation (e.g. standard deviation) or associated estimates of uncertainty (e.g. confidence intervals) |
| ☐ | ☒ | For null hypothesis testing, the test statistic (e.g. *F*, *t*, *r*) with confidence intervals, effect sizes, degrees of freedom and *P* value noted *Give P values as exact values whenever suitable.* |
| ☒ | ☐ | For Bayesian analysis, information on the choice of priors and Markov chain Monte Carlo settings |
| ☒ | ☐ | For hierarchical and complex designs, identification of the appropriate level for tests and full reporting of outcomes |
| ☒ | ☐ | Estimates of effect sizes (e.g. Cohen's *d*, Pearson's *r*), indicating how they were calculated |

*Our web collection on statistics for biologists contains articles on many of the points above.*

## Software and code

Policy information about availability of computer code

| | |
|---|---|
| Data collection | Cryo-ET data were collected on Titan Krios (Thermo Fisher) by an automated EM data acquisition software SerialEM 3.8. |
| Data analysis | Tomograms were motion-corrected using Motioncorr2, aligned and reconstituted by IMOD 4.11. Subtomogram averaging analyses was performed with I3 0.9. The structures were visualized and segmented in the UCSF ChimeraX 1.3. Local resolutions of the structures were measured by Resmap 1.1.4. Molecular Dynamics Flexible Fittings (MDFF) were performed in NAMD 3.0 Alpha with CHARMM36 force-field parameters. Custom scripts for multiple distance spatial cluster analysis, kernel density heatmaps and histogram profiles were generated with Python 3.8. |

For manuscripts utilizing custom algorithms or software that are central to the research but not yet described in published literature, software must be made available to editors and reviewers. We strongly encourage code deposition in a community repository (e.g. GitHub). See the Nature Portfolio guidelines for submitting code & software for further information.

## Data

Policy information about availability of data

All manuscripts must include a data availability statement. This statement should provide the following information, where applicable:
- Accession codes, unique identifiers, or web links for publicly available datasets
- A description of any restrictions on data availability
- For clinical datasets or third party data, please ensure that the statement adheres to our policy

TThe cryo-EM structures have been deposited to the Electron Microscopy Data Bank (EMDB). Cryo-ET structural maps for HIV-1 Env bound to one, two and three CD4 molecules, the consensus structure of HIV-1 Env bound to CD4 receptors in membranes and the unliganded Env trimer have been deposited with accession codes EMD-29292, EMD-29293 and EMD-29294, EMD-29295 and EMD-41045, respectively.

## Human research participants

Policy information about studies involving human research participants and Sex and Gender in Research.

| Reporting on sex and gender | n/a |
| Population characteristics | n/a |
| Recruitment | n/a |
| Ethics oversight | n/a |

Note that full information on the approval of the study protocol must also be provided in the manuscript.

# Field-specific reporting

Please select the one below that is the best fit for your research. If you are not sure, read the appropriate sections before making your selection.

☒ Life sciences    ☐ Behavioural & social sciences    ☐ Ecological, evolutionary & environmental sciences

For a reference copy of the document with all sections, see nature.com/documents/nr-reporting-summary-flat.pdf

# Life sciences study design

All studies must disclose on these points even when the disclosure is negative.

| Sample size | 168 tomograms containing 857 membrane-membrane interfaces were captured in available electron microscopy time. Subtomogram averaging structures from 5712 particles reached the resolution beyond 20 Å to support the conclusions in the manuscript. |
| Data exclusions | No data were excluded from analyses. |
| Replication | Different grids from distinct sample preparations were imaged. |
| Randomization | Membrane-membrane interfaces between HIV-1 and MLV particles were randomly formed and picked with no bias. Structure resolutions were determined by two random halves of the data set. |
| Blinding | The investigators were not blinded during data collection or during analysis. Groups are not relevant to this work. |

# Reporting for specific materials, systems and methods

We require information from authors about some types of materials, experimental systems and methods used in many studies. Here, indicate whether each material, system or method listed is relevant to your study. If you are not sure if a list item applies to your research, read the appropriate section before selecting a response.

## Materials & experimental systems

| n/a | Involved in the study |
|-----|----------------------|
| ☐ | ☒ Antibodies |
| ☐ | ☒ Eukaryotic cell lines |
| ☒ | ☐ Palaeontology and archaeology |
| ☒ | ☐ Animals and other organisms |
| ☒ | ☐ Clinical data |
| ☒ | ☐ Dual use research of concern |

## Methods

| n/a | Involved in the study |
|-----|----------------------|
| ☒ | ☐ ChIP-seq |
| ☒ | ☐ Flow cytometry |
| ☒ | ☐ MRI-based neuroimaging |

## Antibodies

| | |
|---|---|
| Antibodies used | 2G12 (ARP-1476, Lot 140426), Human anti-HIV-1 Neutralizing Serum (ARP-1983, Lot 4/2/91), HRP Goat Anti-Human Ig (Cat. No. 2010-05, Lot D7511-M985C, SouthernBiotech) |
| Validation | Human monoclonal antibody 2G12<br>Antibody class: IgG1κ<br>ARP-1476 is a recombinant monoclonal antibody to HIV-1 gp120.<br>This antibody was produced in a recombinant Chinese Hamster Ovary (CHO) cell expression system and purified by Protein A affinity chromatography. This antibody originates from an HIV-1 positive human donor. This antibody neutralizes a broad variety of SHIV variants, HIV-1 laboratory strains and primary isolates. The epitope is conformational and carbohydrate dependent. It is directed against N-linked glycans in the C2, C3, V4, and C4 domains of gp120.<br><br>Human anti-HIV-1 Neutralizing Serum<br>Host: Human<br>ARP-1983 was pooled from four separate bleeds from the same human immunodeficiency virus type 1 (HIV-1)-positive patient. The antiserum was diluted with PBS and distilled water prior to lyophilization. ARP-1983 is at a 1:2 final dilution of the original sample. ARP-1983 is provided as 1 ampule of sterile, heat inactivated, lyophilized antiserum.<br><br>HRP Goat Anti-Human Ig<br>Isotype: Goat IgG<br>Isotype Control: 0109-05<br>Specificity: Reacts with the heavy and light chains of human IgG, IgM, and IgA<br>Source: Pooled antisera from goats hyperimmunized with human IgG, IgM, and IgA<br>Cross Adsorption: None; may react with immunoglobulins from other species<br>Purification Method: Affinity chromatography on pooled human immunoglobulins covalently linked to agarose<br>Conjugate: HRP (Horseradish Peroxidase) |

## Eukaryotic cell lines

Policy information about cell lines and Sex and Gender in Research

| | |
|---|---|
| Cell line source(s) | HEK293T cells (ATCC)<br>TZM-bl (from NIBSC, Centre For AIDS Reagents, ARP5011)<br>HIV-1BaL/SUPT1-CCR5 cells (cell line ID #CLN204, generated by Jeffrey Lifson laboratory, AIDS and Cancer Virus Program, Frederick National Laboratory for Cancer Research, Frederick, MD, USA)<br>ADA.CM.755*/HEK293T (generated by Michael Zwick laboratory, Department of Immunology and Microbiology, The Scripps Research Institute, La Jolla, CA, USA) |
| Authentication | Virus HIV-1BaL were generated from HIV-1BaL/SUPT1-CCR5 cells. Virus HIV-1ADA.CM.755* were generated from ADA.CM.755*/HEK293T by transfecting HIV-1 gagpol plasmid. MLV-CD4 viral particles were generated from HEK293T cells by co-transfecting MLV gagpol and CD4 receptor expressing plasmid. Plasma membrane blebs were produced from TZM-bl cell line incubated with blebbing buffer. |
| Mycoplasma contamination | The cell lines were not contaminated by mycoplasma. |
| Commonly misidentified lines<br>(See ICLAC register) | None |

