## [Peer Review File · Nature]

Manuscript Title: HIV-1 Env trimers asymmetrically engage CD4 receptors in membranes

Redactions – unpublished data

Reviewer Comments & Author Rebuttals

Reviewer Reports on the Initial Version:

Referees' comments:

Referee #1 (Remarks to the Author):

In their manuscript, Li and co-workers use CryoEM to elucidate the existence of asymmetric HIV-1 Env trimers engaged with one, two and three CD4 molecules during membrane binding. The results provided by the authors are quite exciting and the hypotheses made during the report are supported by the data. There are, however, a number of questions I would like to ask prior to recommend the manuscript suitable for publication in Nature.

Main points:

- Is the information provided new? Over the past few years several reports have shown the step-wise Env reaction and how different Env engage with CD4 forming asymmetric complexes . Even if the structures provided were recovered in a surrogate system where Env is embedded in the native virion engaged with a surrogate target (MLV pseudoviruses expressing CD4), I am not sure that the new data is something that was unknown or missing in our understanding of HIV-1 fusion. Please comment.
- How come the authors do not discuss on the importance of HIV-1 Env prior CD4 engagement (State 1)? It seems that the use of AT-2 not only eliminates retroviral infectivity, which questions the physiological relevance of the results, but also alters the state1 conformation [as previously published by the authors in XXX]. Would it be possible that this disruption stabilizing the Env (in a similar way as then Env SOSIP] also alters the conformational dynamics of Env and therefore their stoichiometry? The authors should discuss this in their report citing their own articles on the matter and other literature. They assume this distortion in Env during sample preparation will not have an impact when interacting with CD4, therefore this Hypothesis should be discussed and validated.
- One of the main hypotheses of the paper is that engagement between HIV-1 Env (assuming that AT-2 treatment does not impact Env dynamics) with MLV pseudoviruses decorated with CD4 resembles the one between functional HIV-1 with target cells (T cells or Macrophages). Nothing is discussed regarding the differences in membrane curvature, CD4 accessibility, CD4 concentration and clustering and other issues regarding membrane composition, cholesterol concentration or SERINC incorporation...why? Please comment and discuss.
- Another aspect regarding MLV as a surrogate for a target T cell, is maturation. The authors, with

great interest and detail, discuss the importance of HIV-1 maturation (see below) and the implications of this on Env clustering; but they have not studied or discussed how MLV maturation might affect CD4 clustering and the implications of this when analyzing Env – CD4 interactions.

- In regards to Env clusters and their role during HIV-1 fusion:

- 1- What is a cluster? The authors should define what they consider to be a cluster. For instance, the relationship between maturation and Env clustering has been seen with STED microscopy [Chojnacki et al., 2013] and also with FRET-FLIM [Carlón-Andrés et al., 2021]. In the former Env clusters were seen with a resolution of around 40 nm utilizing super-resolution light microscopy, in the later Env clusters were defined as inter-molecular FRET with a resolution between Env trimers of around 10 nm. In the present manuscript, clusters seem to be organized only when Env engages with CD4 as a prefusion complex. Moreover, Figure 2a-b shows clusters are structured in rings with a number of Envs that oscillates between 5-15 Env approx. The definition of a cluster, in this case should comprise a wide range of Env-CD4 complexes. In line with this question, can the authors examine their micrographs and check how many clusters of Env could be found in mature viruses (not necessarily engaged with CD4) and compare this with immature particles? This information would be very important as the two reports previously cited assume that Env clusters could be important prior to CD4 interactions. Moreover, in Carlón-Andrés et al., 2021 it was shown that the clusters (as defined by the authors on the basis of the inter-molecular interactions) break apart upon ligand interactions (in all cases sCD4, bNAbs and even T Cells) contributed to dissolve Env association only in mature viruses. Please, cite these reports and discuss.

- Is the number of Env incorporated per virus representative of physiological relevant strains (Tier1 and Tier2) viruses?

- "The dependence on Env clustering can be teste using fusion kinetics", could the authors also show in their fusion experiments HIV(BaL) and compare with JRFL and NL4-3? Functional fusion assays should be carried out with the same viruses imaged by CryoEM (especially because it seems that treatment with AT-2 does not interfere with their tumorigenicity, as stated in Material and Methods).

- The authors assume that delayed fusion kinetics when comparing Env (NL-4-3) versus Env (JR-FL) is due Env clustering (I guess they mean Env – CD4 complex clusters). Kinetics could be delayed by many other factors, for instance tropism, use of co-receptor, differences in stoichiometry, differences in Env dynamics (as shown by the authors in previous reports), or differences in membrane composition. Also, as explained before these so-called clusters are in contradiction with other reports in which the stoichiometry of Env – CD4 interactions is limited to a few Envs (and in the case of JR-FL only one). They should clarify this contradiction. I am worried that the number of Envs and CD4 and the particular membrane curvature of MLV induces these clusters that might not be representative when HIV-1 engages with T cells or macrophages (which also might be different as in the former it seems HIV-1 fuses in the plasma membrane and in the latter within endosomal compartments).

- The fusion kinetics data "suggests that Env clustering is precluded by interactions between matrix and the cytoplasmic tail of Env. Please cite previous reports where this evidence was first established (Chojnacki et al., 2013; Carlón-Andrés et al., 2021; Nieto-Garai et al., 2020 and Haider et al., 2021) and discuss the current results in relation with those reports.

- It is surprising that fully immature particles are able to fuse. I would like to know also the definition of immature particles. In other reports HIV-1 particles were partially labelled with GFP-Gag and different concentrations of protease inhibitor were tested against the so-called saponin assay [see Greg Melikyan's lab publications on the matter]. Clearly there are different degrees of maturation

(from full Gag-GFP release to partial Gag-GFP release and therefore partial protease cleavage). Can the authors demonstrate that their immature particles, when submitted to the saponin assays were indeed immature (no Gag-GFP release at all) with the given concentrations of inhibitor?

- In perspectives, discussion other structural and functional work has already established potential roles and stoichiometry between Env, CD4 and CXCR4 or CCR5. Please cite these reports and comment.

Referee #2 (Remarks to the Author):

Li et al. have set up a clever system using virus like particles to present the full-length CD4 to study how HIV-1 envelope (Env) trimer on the surface of the BaL virions interacts with the membrane-bound CD4 by cryo-electron tomography (cryo-ET). They find that Env-CD4 complexes form clusters and rings at the membrane-membrane interfaces in a capsid maturation-dependent manner. Further subtomogram averaging and classification have revealed that Env trimers form stable complexes with one, two, or three CD4 receptor molecules, suggesting that asymmetric HIV-1 Env trimers with one or two CD4 molecules bound are detectable intermediates during the viral attachment step. Interestingly, the Env in complex with three CD4s still adopts a partially open state, different from what has been reported from the previous studies using soluble proteins. The manuscript is clearly written and the data are well presented, but there are several issues that need to be addressed.

Major points:

1. Sample preparation is perhaps the most important step for a successful cryo-EM or cryo-ET project. The HIV-1 BaL is one of those early viruses, similar to the so-called “lab-adapted” strains, which are very sensitive to soluble CD4 and antibody neutralization, as well as show spontaneous or CD4-induced gp120 shedding. It is unclear why this particular strain was chosen since its Env could be very heterogeneous and bring additional technical challenges to structural analysis, in particular, for cryo-ET that could not reach high resolution to provide independent validation (possibly because of sample availability, but those clinically relevant tier 2 or tier 3 viruses should be used). Nonetheless, the virus sample should be extensively characterized using more recent reagents in the field, such as new broadly neutralizing antibodies. Is Env fully cleaved? Does CD4 binding lead to gp120 dissociation? It is important to know what reconstructed by cryo-ET are cleaved Env trimers or uncleaved trimers. If the Env trimers on these virions are indeed not very homogeneous, it will undermine one’s confidence in these low-resolution reconstructions by cryo-ET.
2. It is stated that “HIV-1 Env binding to CD4 induces Env clustering and ring formation at membrane-membrane interfaces”, but maturation-induced clustering of Env proteins that does not require CD4 has been reported previously (Chojnacki et al., *Science* 2012 Oct 26;338(6106):524-8). Is CD4 needed or not needed for Env clustering, or there are two different types of clustering? Is it possible that the CD4-dependent clustering is caused by the high density of CD4 on the MLV-CD4 VLPs? Moreover, it is believed that “During maturation, MA is cleaved from the capsid, allowing Env to move laterally on the surface of the viral particle”. While it is a very interesting hypothesis, what have been reported in Qu et al., *Science*. 2021 Aug 6; 373(6555): 700–704 suggest that MA trimer

apparently only rotates locally to form a different hexameric lattice upon maturation. How could Env move laterally if it remains associated with the MA lattice?

3. Comparisons among different virus preparations or particles should be performed more rigorously. For example, to compare fusion kinetics of HIV-1NL4-3 and HIV-1JR-FL strains, it is unclear whether the same number of functional Env trimers get incorporated into one virion for the two strains or whether there are any intrinsic differences in fusion kinetics because of the variations in the Env sequence. There are many possible reasons for the observed differences in fusion kinetics. Likewise, the C-tail deletion of Env could change its incorporation level and other properties that could alter the fusion kinetics, not necessarily just Env clustering.

4. The author's group has reported previously that HIV-1 Env trimers are conformational dynamic and have spontaneous access to open conformational states including the CD4-bound state. How do the asymmetric HIV-1 Env trimers with a single and two CD4 molecules bound in the current study (apparently stable enough) correlate with those FRET states?

5. If the observed CD4-bound Env complexes are stable intermediates for the fusion process, why wouldn't they be targeted by those nonneutralizing CD4i antibodies that target the coreceptor binding site?

Minor points:

1. With low contrast and crowded space between two membranes, how were Env-CD4 complexes at membrane-membrane interfaces picked manually? No sufficient details were given for Subtomogram averaging. How was an initial reference generated for alignment?

2. "Quantification of tomograms revealed that membrane-membrane interfaces with 156 immature capsids had fewer Env trimers (4.6 ± 2.5) than interfaces with mature capsids (7.5 ± 4)". Is the difference between the range of 2.1-7.1 and 3.5-11.5 really significant?

3. "There are considerable steric constraints in the gp120-CD4 interaction that must be overcome for the Env trimer to bind the co-receptor. This could be achieved by either 1) shedding of the gp120-CD4 complex". Is coreceptor binding to dissociated gp120 relevant to membrane fusion?

Referee #3 (Remarks to the Author):

The study uses cryo-ET imaging and subtomogram averaging to investigate the interaction of HIV-1 Env and its receptor CD4 in the context of native membranes. Subtomogram averaging is used to determine low resolution structures of the Env trimer in complex with CD4 including one, two, and three CD4-bound states. The asymmetric one and two bound states have not been observed before and are associated with an opening of the trimer based on comparison to previously described closed (unliganded) and open (3 bound) structures. Subsequent co-receptor engagement and membrane fusion requires additional structural changes.

This work describes a novel and well-designed system to observe interactions between Env and CD4 on the membranes of HIV-1 Bal particles and MLV-CD4 VLPs which cluster in the membrane at the

interface between the two types of particles. Using cryo-EM imaging, the ms reveals structural changes that occur when Env interacts with CD4 receptors. Clustering is dependent on the maturation state of HIV and is shown to depend on free mobility of cytoplasmic tails. When the co-receptor CCR5 is added along with CD4, membrane fusion occurs, though CCR5 is not part of the structural investigation and the number of trimers involved in membrane fusion is not specifically determined.

The images and structures are extremely interesting and the observation of asymmetric binding and opening of Env and the mode of CD4 interaction with the membrane are important for understanding HIV entry and to extend structural work mostly based on isolated proteins. The main concern is the low resolution of the structures which limits interpretation. Re-arrangement of specific Env structural elements is plausible but not definitive at the level of protomer structure. The one CD4-bound structure has very weak density. A larger dataset by the same approach is desirable and would lead to more robust conclusions and impact.

Specific points:

1. Quantitative assessment of opening or asymmetry should be included and is important in describing the findings.
2. The subtomogram averages appear to include specific non-parallel membranes associated with the interacting molecules, features which may be an important feature of the binding modes in the different types of particle interfaces and requires further analysis and discussion. This is important both to the data analysis and to comparison to other types of fusion systems such as SNAREs.
3. Calculating subtomogram averages for unliganded data may be a valuable control that would demonstrate the quality of averaging and features obtained from more particles and confirm that the unliganded structures are as expected. This should be feasible given that the authors have already identified positions of those particles in their inter-molecule distance measurements.
4. The data processing strategy for the tomography dataset involved classifying the subtomograms based on chosen membrane-membrane distances (method not described in sufficient detail) though the structures may in fact vary continuously with distance. A less biased approach would involve classifying all the subtomograms based on CD4 binding state and then identify their locations in the tomograms. They could also assign the number of different CD4 bound states for the different interaction clusters (loosely defined as small, large and rings in the ms). At present there are relatively few particles for the different categories and all initial chosen particles are retained in the final averages, without rejections based on quality, so improvements are possible.

Referee #4 (Remarks to the Author):

Review for Li et al. MTS-2022-12-20465A

The authors present a concise manuscript outlining the acquisition of structural information of HIV-1's Env complexes in viral context; a comparative analysis to MLV's Env is also derived as part of the present work. The experiments carried out in the present manuscript are certainly of interest to a broad community as there exists a debate about the structural details of HIV Env in infective particles. The work performed by the authors is in the context of infectious particles which positions them in a unique situation compared to previous work. However, the lack of sub-nanometer structures and systematic analysis of the derived models reduces the enthusiasm of this reviewer.

Major concerns

1. The major finding from the manuscript seems the asymmetric nature of Env trimers when bound to CD4 receptors. However the lack of sub nanometer structures in the present work limits the interpretability of the structural data. Furthermore the findings are not supported by virological assays.
2. The authors refer to their models as atomistic; this reviewer begs to differ. No attempt was made to assess the quality of the MDFF-derived models via any standard metrics. For instance, there is not information regarding the structural stability of the resulting models (e.g., root-mean-squared differences traces, model-to-map fits, etc). There is not assessment on the cross correlation between model and density. Certainly, the maps could have been used in a more rigorous way to produce higher quality models.
2. The analysis pertaining to the models derived from the densities is ambiguous and not very rigorous (see point #2). The authors refer to an open, closed and "more open" conformation. The ambiguity is not only in the language, but also the authors should define unbiased metrics that allow them to classify the models unambiguously.
3. As with any manual selection of VLPs there is the possibility of introducing a human-bias. Although the authors are experts in the field of cryoET, it is the view of this reviewers that there should be an attempt to convince readers that the results are invariant to particle selection.

Minor concerns

1. The MDFF simulations calculations were performed 'in vacuo'; these types of calculations are acceptable for specialized journals. At the level of Nature, this reviewer would expect that the derivation of full-length models in realistic lipid membranes would be the norm.

Author Rebuttals to Initial Comments:

Response:

To the Editor and Reviewers,

We appreciate the opportunity to respond to reviewer's comments and concerns.

We would first like to emphasize that we are aware of the limitations in the resolutions of our structures. In structural biology there is often a tradeoff between reaching higher resolution and gaining insights into interesting biology in heterogenic systems. In this work we chose to prioritize gaining insights into biology over reaching high resolution structures. By doing so, we were able to reveal that HIV-1 Env binds one, then two, and finally three CD4 molecules with the Env trimers adopting asymmetric conformational states in each of these bound states. The insights we achieved would have been impossible if we had only aimed for higher resolution by limiting our focus to homogeneous objects. We solved the structure of the unliganded Env trimer to ~1 nm resolution from the same data set as Reviewer 3 suggested. This illustrates that the problem is the heterogeneity in Env-CD4 complexes and not sample quality or subtomogram averaging and classification that can reach ~1 nm resolution for more homogeneous objects such as the unliganded Env trimer in the same data set. We also believe that the concerns raised by reviewers about low resolution are in part addressed by the accompanying paper by Dam, Bjorkman et al. that provides high-resolution structures for soluble HIV-1 trimers bound to one and two CD4 receptor molecules. The structure of the open trimer bound to two CD4 receptor molecules is consistent with the density observed in our membrane-membrane interfaces. The main structure of the trimer bound to one CD4 molecule was a closed trimer for Dam et al., but an open trimer in our biological membranes. Interestingly, a careful re-analysis by Dam et al. revealed that one third of the BG505 HT1 trimers bound to CD4 are open trimers with mostly one CD4 bound. The revised manuscript by Dam et al. includes this additional analysis. The open trimer bound to one CD4 molecule fits well into our cryo-ET map of an open trimer with one CD4 molecule bound. Moreover, the soluble BG505 trimers bound to one CD4 were engineered to only bind a single CD4 molecule. Thus, a single CD4 is able to open a subset of Env trimers also for the soluble BG505 trimers. The two manuscripts are thus consistent, particularly as it is expected that a more CD4-resistant Tier 2 HIV-1 isolate such as BG505 would be less responsive to CD4 than the Tier 1b isolate BaL used for cryo-ET.

We believe that this is a case where two papers (ours and Dam et al.) using independent scientific paths of exploration arrive at similar conclusions while also complementing each other by addressing the other's weaknesses. We hope that the reviewers were allowed to review the two manuscripts together. Due to heterogeneity, our manuscript using cryo-electron tomography of complete virus particles cannot reach

higher resolution without compromising insights into the biology. In contrast, the cryo-EM used by Dam *et al.* reaches high resolution with molecular insights into the trimer structure, but importantly has to rely on engineered asymmetric trimers without knowing whether they exist in biological membranes. Our report documents that these asymmetric trimers do exist in physiological membranes. Neither paper is as strong independently as they are together; together both studies represent a more complete analysis of the molecular mechanism of HIV-1 Env binding to CD4 receptor across multiple scales of resolution.

We like to highlight one additional important change in the revised manuscript. We believe Reviewer 1 is right in questioning how direct the observed Env clustering at membrane-membrane interfaces correlate with fusion kinetics given that this is a complex multi-factorial system. We have therefore decided to remove the fusion kinetic analyses. These soft correlations distracted from the structural work.

Please find a point-by-point response to reviewers concerns below.

Referees' comments:

Referee #1 (Remarks to the Author):

In their manuscript, Li and co-workers use CryoEM to elucidate the existence of asymmetric HIV-1 Env trimers engaged with one, two and three CD4 molecules during membrane binding. The results provided by the authors are quite exciting and the hypotheses made during the report are supported by the data. There are, however, a number of questions I would like to ask prior to recommend the manuscript suitable for publication in Nature.

Main points:

- Is the information provided new? Over the past few years several reports have shown the step-wise Env reaction and how different Env engage with CD4 forming asymmetric complexes. Even if the structures provided were recovered in a surrogate system where Env is embedded in the native virion engaged with a surrogate target (MLV pseudoviruses expressing CD4), I am not sure that the new data is something that was unknown or missing in our understanding of HIV-1 fusion. Please comment.

Response: This is the first direct visualization of the stepwise binding of HIV-1 Env trimers to CD4 molecules in biological membranes and it reveals asymmetric intermediates of Env in which individual protomers adopt distinct conformational states. Previous evidence of this was indirect, being based on fluorescence imaging, and alternative interpretations of data are possible. Moreover, previous smFRET experiments in our lab (Ma *et al.* 2018, PMID: 29561264) as well as the accompanying manuscript by Dam, Bjorkman and colleagues deliberately engineered asymmetric trimers to observe the binding of one or two CD4 receptor molecules. However, there has been no data to support that engineered asymmetric trimers are good representatives of wild-type HIV-1 Env. The report presented here by Li and colleagues answers this open question by demonstrating that asymmetric Env trimers bound to one or two CD4 receptor molecules exist in biological membranes. Additionally, the importance of asymmetry remains greatly underappreciated in biology. The human brain is biased towards symmetry. EM methods favor symmetry and applying symmetry increases resolution. Thus, many published HIV-1 Env structures are biased towards symmetry. Here we have neither enforced symmetry nor have we introduced reference bias into our structures in an attempt to increase resolution. This results in lower resolution structures, but has allowed us to reveal asymmetric intermediates in a biologically relevant experimental system to study early steps of HIV-1 Env binding to CD4 receptor.

- How come the authors do not discuss on the importance of HIV-1 Env prior CD4 engagement (State 1)? It seems that the use of AT-2 not only eliminates retroviral infectivity, which questions the physiological relevance of the results, but also alters the state1 conformation [as previously published by the authors in XXX]. Would it be possible that this disruption stabilizing the Env (in a similar way as then Env SOSIP) also alters the conformational dynamics of Env and therefore their stoichiometry? The authors should discuss this in their report citing their own articles on the matter and other literature. They assume this distortion in Env during sample preparation will not have an impact when interacting with CD4, therefore this Hypothesis should be discussed and validated.

Response: We agree and discuss this unresolved question about an additional conformational state identified by smFRET in the discussion on pages 16&17. Specifically, applying smFRET to sample the conformational dynamics of HIV-1 Env, we have described a conformational state designated State 1 for unliganded Env on the surface of viruses (Munro 2014, Lu 2019, PMID: 25298114 and 30971821) that is affected upon treatment with AT-2 (Li 2020, PMID: 32601441). While solving the structure of State 1 is a priority for our lab, we do not believe that it affects the results reported in this manuscript because it is upstream of interactions with receptor. Importantly, AT-2 treated viruses are on pathway. AT-2 treatment of HIV-1 does not affect the fusogenicity of HIV-1 Env. This has been well established by Julian Bess and Jeff Lifson (PMID: 9733838) and shown again more recently by Kelly Lee and colleagues (PMID: 35123651). We used a newly plate reader permitting real-time monitoring of luciferase activity to measure the effects of AT-2 on HIV-1 fusion, and in our hands, AT-2 treatment does not alter the fusion kinetics of JR-FL and only slightly affects that of NL4-3.

Fusion kinetics of HIV-1 VLPs carrying Env_{JR-FL} (Black) and Env_{NL4-3} (Red), comparing to the VLPs generated from same batch but treated with AT-2, Env_{JR-FL} (Purple) and Env_{NL4-3} (Pink). Notably, the plate reader used here has been updated, allowing for continuous real-time monitoring of the fusion progress. In this fusion kinetics analysis, Env_{NL4-3} still showed a lag at the initial stage, indicating a delay in the fusion process as originally observed. Importantly, treatment with AT-2 did not have a significant impact on the fusion kinetics of Env_{JR-FL} and only moderately affected that of Env_{NL4-3}.

We have explored various alternative methods for virus inactivation over the years (unpublished), and unfortunately, they have shown conformational and morphological effects that are even worse than those observed with AT-2. For the time being, AT-2 treated viruses prepared from chronically infected cells remain the best source for cryoET studies.

Unpublished data in the Mothes lab demonstrates that State 1 is maintained in a cholesterol-rich microdomain. Cholesterol-rich domains are unlikely to support membrane fusion. We hypothesize that Env will have to laterally partition into different lipid domains with higher disorder to be able to interact with CD4 receptor and undergo CD4-induced conformational changes. We believe that State 1 is upstream of CD4 binding and therefore not a concern for this work on CD4 binding and Env-CD4 complex clustering at membrane-membrane interfaces. Solving the structure of State 1 will require a multi-pronged approach that bypasses the use of AT-2 for chemical inactivation.

- One of the main hypotheses of the paper is that engagement between HIV-1 Env (assuming that AT-2 treatment does not impact Env dynamics) with MLV pseudoviruses decorated with CD4 resembles the one between functional HIV-1 with target cells (T cells or Macrophages). Nothing is discussed regarding the differences in membrane curvature, CD4 accessibility, CD4 concentration and clustering and other issues regarding membrane composition, cholesterol concentration or SERINC incorporation...why? Please comment and discuss.

Response: To directly address the reviewer's concern that MLV particles do not represent native membranes, we generated plasma membrane blebs from CD4 expressing TZM-bl cells rather than relying on the use of MLV pseudoviruses. Membrane blebs present CD4 in a native plasma membrane environment. As shown in Figure 1 of the revised manuscript as well as below, the clusters of Env-CD4 complexes are still observed at the membrane-membrane interfaces. This includes large plasma membrane blebs with low membrane curvature. Our findings recapitulate studies with SNAREs that have demonstrated that the phenotypes seen with small liposomes has been reproduced with plasma membrane blebs, most likely because the high curvature of the donor vesicle or virus dominates over the varying degrees of curvature of the target membranes (PMIDs: 22653732; 23240085, 32860428). A

Careful analysis of Env-receptor and coreceptor interactions in plasma membrane using membrane blebs including from T cells requires a future study.

With respect to the virus used in our studies, as above, we believe that State 1 is upstream of CD4 binding. AT-2 treated viruses show no defect in the fusogenicity of HIV-1 Env and are on pathway. With respect to SERINC, we use native HIV-1BaL containing Nef that will downregulate SERINC and exclude SERINC from incorporation into virus particles. We like to note that we discussed the likely importance of membrane bending on the CD4 receptor side to allow the Env trimer to engage coreceptor.

Thus, collectively, we believe that the study of HIV-1 Env – CD4 receptor interactions in biological membranes using viruses and VLPs represents an advance, allowing new insights over previous studies with soluble ligands and indirect studies based on fluorescence. The reviewer is likely right that MLV particles do not support efficient fusion but this allowed us to accumulate nearly 6000 Env-CD4 complexes in membrane-membrane interfaces. Since fusion is a rapid process, characterizing intermediates requires a slowing down of the fusion reaction permitting the accumulation of enough molecules in an intermediate for structural analysis.

Last but not least, the revised manuscript doesn't over-emphasize the importance of Env clustering, doesn't propose a correlation between Env clustering and fusion kinetics as we have removed the fusion kinetics from the manuscript.

- Another aspect regarding MLV as a surrogate for a target T cell, is maturation. The authors, with great interest and detail, discuss the importance of HIV-1 maturation (see below) and the implications of this on Env clustering; but they have not studied or discussed how MLV maturation might affect CD4 clustering and the implications of this when analyzing Env – CD4 interactions.

Response: HIV-1 Env is known to interact with matrix through the cytoplasmic tail, which is the basis for HIV-1 maturation's effects on Env clustering. We are not aware that CD4 receptor has a direct interaction with MLV capsid. Moreover, as shown above and presented in Figure 1 of the revised manuscript, we have observed Env-CD4 clusters in native plasma membrane blebs.

- In regards to Env clusters and their role during HIV-1 fusion:

- 1- What is a cluster? The authors should define what they consider to be a cluster. For instance, the relationship between maturation and Env clustering has been seen with STED microscopy [Chojnacki et al., 2013] and also with FRET-FLIM [Carlon-Andres et al., 2021]. In the former Env clusters were seen with a resolution of around 40 nm utilizing super-resolution light microscopy, in the later Env clusters were defined as inter-molecular FRET with a resolution between Env trimers of around 10 nm. In the present manuscript, clusters seem to be organized only when Env engages with CD4 as a prefusion complex. Moreover, Figure 2a-b shows clusters are structured in rings with a number of Envs that oscillates between 5-15 Env approx. The definition of a cluster, in this case should comprise a wide range of Env-CD4 complexes. In line with this question, can the authors examine their micrographs and check how many clusters of Env could be found in mature viruses (not necessarily engaged with CD4) and compare this with immature particles? This information would be very important as the two reports previously cited assume that Env clusters could be important prior to CD4 interactions. Moreover, in Carlon-Andres et al., 2021 it was shown that the clusters (as defined by the authors on the basis of the inter-molecular interactions) break apart upon ligand interactions (in all cases sCD4, bNAbs and even T Cells) contributed to dissolve Env association only in mature viruses. Please, cite these reports and discuss.

Response: We appreciate the careful questioning by Reviewer 1 here and in the subsequent points about the physiological relevance of the Env clustering observed at membrane-membrane interfaces. The correlation between observed clustering of Env and a delay in the fusion kinetics is soft, the underlying biology multi-factorial, and requires more work to be convincing. Importantly, this part of the manuscript distracts from the structural analysis of Env-CD4 receptor complexes. Therefore, we have decided to remove the fusion kinetic analyses. We report Env clustering as an obvious phenotype that requires reporting but are no longer attempting to imply that the Env clustering is biologically important for fusion. With respect to a better definition of Env clusters, we have performed multiple distance spatial cluster analysis. Unexpectedly, we find that the Env clustering doesn't significantly change between immature and mature viruses at smaller distances. Instead, clear clustering is observed on mature particles when CD4 is present at the membrane-membrane interfaces, while no clustering is observed on immature particles. However, we do observe that immature particles display a peak at large search ranges. This indicates that Env trimers on immature particles are near perfectly dispersed compared to those on mature particles. Thus, the previous increase in clustering seen in mature particles is visible in our analyses as a near perfectly dispersed Env distribution in immature virus particles, a phenotype that is lost upon maturation. We included this analysis in Fig. 2i of the revised manuscript and discuss it in the text. Thus, while different experimental settings likely explain the different cluster sizes in our system compared to previous studies, we also see a redistribution of dispersed Env towards clusters following maturation.

- Is the number of Env incorporated per virus representative of physiological relevant strains (Tier1 and Tier2) viruses?

Response: Most people don't realize that there is very little generalizable data on the number of Env trimers incorporated into HIV-1 particles because most cells that are used to produce HIV-1 in tissue culture do not efficiently incorporate HIV-1 Env. For this reason, the literature on Env numbers is severely compromised. The HIV-1 preparations from the chronically infected SupT1/CCR5 cells are one of the best sources of HIV-1. They also represent the best HIV-1 samples for structural studies.

- "The dependence on Env clustering can be teste[d] using fusion kinetics", could the authors also show in their fusion experiments HIV(BaL) and compare with JRFL and NL4-3? Functional fusion assays should be carried out with the same viruses imaged by CryoEM (especially because it seems that treatment with AT-2 does not interfere with their tumorigenicity [sic; fusogenicity intended?], as stated in Material and Methods).

Response: We believe that the reviewer is right with his/her/their questions about the weak correlation between fusion kinetics and Env clustering. Therefore, we have removed the fusion kinetics.

- The authors assume that delayed fusion kinetics when comparing Env (NL-4-3) versus Env (JR-FL) is due Env clustering (I guess they mean Env – CD4 complex clusters). Kinetics could be delayed by many other factors, for instance tropism, use of co-receptor, differences in stoichiometry, differences in Env

dynamics (as shown by the authors in previous reports), or differences in membrane composition. Also, as explained before these so-called clusters are in contradiction with other reports in which the stoichiometry of Env – CD4 interactions is limited to a few Envs (and in the case of JR-FL only one). They should clarify this contradiction.

Response: As above, we agree with the reviewer that Env clustering may not be the only explanation for our results and have removed the fusion kinetics. We like to note that there has never been any parallel EM work for the virus preparations studied in the Env stoichiometry field by Trkola, Sodroski, and others. The entirety of Env stoichiometry work is based on weak assumptions. In fact, the Env trimer numbers that are used for the computational modeling are all based on the very same HIV-1BaL preparations studied here. However, all experimental data is based on viruses produced in HEK293 cells, where most particles, on average, don't even carry any single Env trimer. These reports need to be validated using parallel cryo-ET to actually count the number of Env trimers per virus and determine the distribution of Env trimers per particle in the virus preparations used to perform infectivity experiments.

I am worried that the number of Envs and CD4 and the particular membrane curvature of MLV induces these clusters that might not be representative when HIV-1 engages with T cells or macrophages (which also might be different as in the former it seems HIV-1 fuses in the plasma membrane and in the latter within endosomal compartments).

Response: We greatly de-emphasized the importance of Env clustering in the revised manuscript. It is an obvious phenotype that we believe should be quantified and reported in a single figure, but not be over-interpreted. By de-emphasizing the physiological importance of Env clustering, we hope that the readers realize that structural analysis based on subtomogram averaging is only possible if experimental conditions are identified where thousands of Env-CD4 complexes accumulate at membrane-membrane interfaces.

We do believe that the Env clusters should be reported since they are still observed at HIV-1BaL- plasma membrane bleb interfaces, including with very large blebs (as shown above). Thus, curvature is not the sole determinant of Env clustering. For SNAREs, experiments for plasma membrane blebs reproduced the findings observed in small liposomes (PMIDs: 22653732; 23240085, 32860428). Additionally, though fluorescence-based imaging in living cells performed by Sergi Padilla-Parra does not support the formation of large Env clusters (PMID: 34707229), the imaging was performed with large GFP insertions into the Env ectodomain and bound atto-fluorophore conjugated nanobodies. These bulky insertions and additions may compromise higher order Env-Env and Env-receptor interactions. Every experimental approach introduces some caveats. Parallel cryoET-fluorescence studies will be needed to resolve this.

- The fusion kinetics data "suggests that Env clustering is precluded by interactions between matrix and the cytoplasmic tail of Env. Please cite previous reports where this evidence was first established (Chojnacki et al., 2013; Carlon-Andres et al., 2021; Nieto-Garai et al., 2020 and Haider et al., 2021) and discuss the current results in relation with those reports.

Response: We agree and cite these reports. Note that the revised manuscript does not contain fusion kinetic experiments.

- It is surprising that fully immature particles are able to fuse. I would like to know also the definition of immature particles. In other reports HIV-1 particles were partially labelled with GFP-Gag and different concentrations of protease inhibitor were tested against the so-called saponin assay [see Greg Melikyan's lab publications on the matter]. Clearly there are different degrees of maturation (from full Gag-GFP release to partial Gag-GFP release and therefore partial protease cleavage). Can the authors demonstrate that their immature particles, when submitted to the saponin assays were indeed immature (no Gag-GFP release at all) with the given concentrations of inhibitor?

Response: We appreciate the reviewer's feedback. As mentioned above, given the many concerns, we have decided to remove the fusion kinetic experiments and will assemble a more complete future manuscript along the reviewer's recommendations. Note, with respect to EM data, since we directly observed the viral particles under EM, we could clearly see the sphere-shaped capsids that are a classic feature of immature particles. About 99% particles we saw in our tomograms were immature particles. Consistent with the observation by saponin assay from Greg Melikyan's lab, a few immature particles

showed discontinuous shell, suggesting partial protease cleavage. However, most membrane-membrane interfaces we observed were generated at the membranes on top of the intact sphere-shaped capsid.

- In perspectives, discussion other structural and functional work has already established potential roles and stoichiometry between Env, CD4 and CXCR4 or CCR5. Please cite these reports and comment.

Response: We discuss these findings in the introduction (pages 4 to 5 transition). As mentioned above, there is no overinterpretation of Env clustering in the revised manuscript and the fusion kinetics have been removed.

Referee #2 (Remarks to the Author):

Li et al. have set up a clever system using virus like particles to present the full-length CD4 to study how HIV-1 envelope (Env) trimer on the surface of the BaL virions interacts with the membrane-bound CD4 by cryo-electron tomography (cryo-ET). They find that Env-CD4 complexes form clusters and rings at the membrane-membrane interfaces in a capsid maturation-dependent manner. Further subtomogram averaging and classification have revealed that Env trimers form stable complexes with one, two, or three CD4 receptor molecules, suggesting that asymmetric HIV-1 Env trimers with one or two CD4 molecules bound are detectable intermediates during the viral attachment step. Interestingly, the Env in complex with three CD4s still adopts a partially open state, different from what has been reported from the previous studies using soluble proteins. The manuscript is clearly written and the data are well presented, but there are several issues that need to be addressed.

Major points:

1. Sample preparation is perhaps the most important step for a successful cryo-EM or cryo-ET project. The HIV-1 BaL is one of those early viruses, similar to the so-called “lab-adapted” strains, which are very sensitive to soluble CD4 and antibody neutralization, as well as show spontaneous or CD4-induced gp120 shedding. It is unclear why this particular strain was chosen since its Env could be very heterogenous and bring additional technical challenges to structural analysis, in particular, for cryo-ET that could not reach high resolution to provide independent validation (possibly because of sample availability, but those clinically relevant tier 2 or tier 3 viruses should be used). Nonetheless, the virus sample should be extensively characterized using more recent reagents in the field, such as new broadly neutralizing antibodies. Is Env fully cleaved? Does CD4 binding lead to gp120 dissociation? It is important to know what reconstructed by cryo-ET are cleaved Env trimers or uncleaved trimers. If the Env trimers on these virions are indeed not very homogenous, it will undermine one’s confidence in these low-resolution reconstructions by cryo-ET.

Response: We agree with the reviewer that sample preparation is key, and emphasize the HIV-1BaL preparations at the beginning of the result section. It turns out that there are only two cell lines that generate HIV-1 preparations of a quality suitable for cryo-ET. This is because HIV-1 particles generated in commonly used fibroblast cells produce mostly bald particles with no Env trimers on their surfaces. The infectivity of these preparations is high, but most particles are unusable at the EM level as very few particles (less than 1%) actually incorporate Env trimers. Because there are so few Env trimers, subtomogram averaging of these virus preparations is not possible. As the reviewer also points out, many cells do not process Env into cleaved mature Env as efficiently as T-cells. Julian Bess and Jeff Lifson addressed this problem by chronically infecting SupT1 cells stably expressing CCR5 with HIV-1BaL and then selecting for stable virus production. These cells generate high-quality virus particles. In fact, almost all cryo-ET studies, including those by Sriram Subramaniam and many other groups, have relied on these cells (PMIDs: 16728975, 18668044, 21203482, 23267106, 25569620, 32601441). Julian Bess and Jeff Lifson have tried to apply this method to numerous HIV-1 isolates, but they have only succeeded very few times. To address the issue of Env processing, it is known that the Env from this cell line is fully cleaved.

The HIV-1BaL used here is a primary HIV-1 isolate (PMID: 3014648). The lab-adapted versions, which are not used here, are BaL.1 and BaL.26. HIV-1BaL is a Tier1b virus, meaning that it is more open and more CD4 sensitive than Tier 2 and 3 isolates, but it is not as open as lab-adapted HIV-1 isolates such as NL4-3.

[Western blot (REDACTED)].

The second cell line that generates viruses suitable for cryoET was generated by Michael Zwick at Scripps (PMID: 28446665). An HIV-1ADA Env expressing cell line was deliberately selected for high levels of Env on the plasma membrane and high Env incorporation into virus particles. This is a cell line stably expressing Env and particles are generated by transfection of plasmids encoding GagPol. Virus particles generated from this cell contain up to ~150 Env trimers on their surfaces. It is known that during the selection process, a large portion of the cytoplasmic tail of Env was lost. We have included data using particles generated from these cells in this manuscript. They have allowed us to increase the number of observable Env-CD4 complexes. We have not observed a difference in the way EnvBaL and EnvADA approach CD4, so the data sets from both virus preparations were combined. However, because there can be ~150 per particle and because the deletion of the C-tail of Env that includes the region determined to be essential for the interaction with matrix, the clustering analysis has been performed with the more physiologically representative HIV-1_{BaL}.

Without these two sources of HIV-1 viruses suitable for cryo-ET studies, this work would not have been possible. To acknowledge this, Julian Bess, Jeff Lifson, and Michael Zwick are co-authors on this report.

2. It is stated that “HIV-1 Env binding to CD4 induces Env clustering and ring formation at membrane-membrane interfaces”, but maturation-induced clustering of Env proteins that does not require CD4 has been reported previously (Chojnacki et al., Science 2012 Oct 26;338(6106):524-8). Is CD4 needed or not needed for Env clustering, or there are two different types of clustering? Is it possible that the CD4-dependent clustering is caused by the high density of CD4 on the MLV-CD4 VLPs? Moreover, it is believed that “During maturation, MA is cleaved from the capsid, allowing Env to move laterally on the surface of the viral particle”. While it is a very interesting hypothesis, what have been reported in Qu et al., Science. 2021 Aug 6; 373(6555): 700–704 suggest that MA trimer apparently only rotates locally to form a different hexameric lattice upon maturation. How could Env move laterally if it remains associated with the MA lattice?

Response: As mentioned in response to Reviewer 1, we have also performed multiple distance spatial cluster analyses and are finding that Env clustering is clearly induced by CD4 at membrane-membrane interfaces (Fig. 2g and h in revised manuscript). When comparing immature and mature virus particles, while we did not observe a pronounced enhancement for small clusters in mature particles, we do see a peak at large search ranges in immature particles. This indicates that Env trimers on immature particles are near perfectly dispersed compared to those on mature particles. Thus, the previous increase in clustering seen in mature particles is visible in our analyses as a near perfectly dispersed Env distribution in immature virus particles that disappears following maturation.

With respect to Env mobility, several studies have reported an increase in Env mobility following maturation. Qu *et al.* used an engineered GagPol that allowed for a continued association of the mature matrix with the immature capsid in their imaging study. However, in wild type GagPol, matrix is fully cleaved from capsid during maturation. Numerous labs are currently studying the molecular interaction of Env with matrix in immature and mature particles. That said, though the details of the interaction between matrix and Env are currently unknown, it is well established that Env mobility increases upon maturation. Importantly, the revised paper is not trying to claim that Env clustering is required for HIV-1 fusion. We have removed the fusion kinetics because the correlation between clustering and delayed fusion kinetics is soft. Reviewer 1 is right that many other factors could influence the observed differences. We report Env clustering because it is an obvious phenotype and allows us to observe many Env-CD4 receptor complexes permitting subtomogram averaging.

3. Comparisons among different virus preparations or particles should be performed more rigorously. For example, to compare fusion kinetics of HIV-1NL4-3 and HIV-1JR-FL strains, it is unclear whether the same number of functional Env trimers get incorporated into one virion for the two strains or whether there are any intrinsic differences in fusion kinetics because of the variations in the Env sequence. There are many possible reasons for the observed differences in fusion kinetics. Likewise, the C-tail deletion of Env could change its incorporation level and other properties that could alter the fusion kinetics, not necessarily just Env clustering.

Response: We agree and a rigorous study cannot be performed in the time frame set by the parallel manuscript. We have therefore decided to remove the fusion kinetic analyses from the current manuscript and focus on the structural characterization of HIV-1 Env – CD4 receptor interactions.

4. The author's group has reported previously that HIV-1 Env trimers are conformational dynamic and have spontaneous access to open conformational states including the CD4-bound state. How do the asymmetric HIV-1 Env trimers with a single and two CD4 molecules bound in the current study (apparently stable enough) correlate with those FRET states?

Response: smFRET performed in our lab has identified at least 3 conformational states of HIV-1 Env. In this smFRET approach, donor and acceptor fluorophores reside in variable loops 1 and 4. A second perspective was monitored with fluorophores in position 542 in gp41 and variable loop 4. For these two perspectives, we observed 3 main conformational states. From the most populated state, designated State 1, Env opens in response to CD4 interaction by passing through one necessary intermediate, State 2, and into the open State 3 conformation (PMID: 25298114). The assignment of structures to these states revealed that the CD4-bound conformational state corresponds to State 3, but States 2 and particularly State 1 are less well defined. Interestingly, within this experimental system, we hypothesized in 2018 that the necessary intermediate in Env opening arises from asymmetric trimers bound to only one or two CD4 molecules. We engineered trimers that can only bind one or two CD4 molecules (similar to how Dam *et al.* engineered trimers in the parallel report) and observed that the intermediate State 2 FRET state was always next to a protomer bound to CD4, while the protomer bound to CD4 was in the open, CD4-bound State 3 conformation (PMID: 29561264). The cryo-ET structures of asymmetric Env trimers in biological membranes observed in this report recall these early hypotheses based on smFRET (PMID: 29561264). To determine the structure of State 1 goes beyond the scope of our work presented here. We are also intensely working on multi-perspective smFRET imaging to gain a more complete picture of the conformational landscape of Env that should interface better with the numerous different structures that have been solved for soluble trimers. These two manuscripts alone describe four trimer structures featuring distinct conformational states of individual protomers: 1) A closed Env trimer bound to one CD4 receptor molecule (Dam *et al.*); 2) An open Env trimer bound to one CD4 molecule (both manuscripts); 3) An open Env trimer bound to two CD4 molecules (both manuscripts); and an open Env trimer bound to three CD4 molecules (Li *et al.*, and previous Bjorkman structures). We believe that State 1 is upstream of these conformational states described here. A multi-perspective smFRET imaging approach will be required to better describe the conformational complexity. We have introduced a paragraph in the Discussion to describe this. Reviewer 1 also asked for a clarification.

5. If the observed CD4-bound Env complexes are stable intermediates for the fusion process, why wouldn't they be targeted by those nonneutralizing CD4i antibodies that target the coreceptor binding site?

Response: We appreciate this question as it reveals further insights into the underlying evolution of these intermediates. While these intermediates in our cryo-ET images are clearly open (gp120 moves away from the central axis), the molecular structure for soluble trimers bound to two CD4 molecules solved by Dam and Bjorkman in the parallel manuscript reveals that the V1V2 loops of the unbound protomer remain on top of the trimer apex. Dam *et al.* have designated this an "occluded-open" state. This structure explains at the atomic level why the out-ward projecting V1V2 loops cannot be observed in our cryoET maps on the protomers that don't bind CD4. The Bjorkman lab has identified some antibodies that could target this occluded-open state, but these antibodies are distinct from the classic non-neutralizing CD4-induced antibodies (PMID: 35136084; PMID: 36448805). Another factor that could explain why non-neutralizing antibodies don't bind well to the CD4-bound protomers is that the IgGs are larger as compared to Fab used for structural studies and therefore encounter steric constraints when trying to bind the Env trimer (PMID: 12970440). Together these structural insights unveil a mechanism for the Env trimer opening while still maintaining its immune evasion features because the V1V2 loops remain at the apex preventing the binding of non-neutralizing antibodies to the opening trimer. We now mention this in the discussion on page 17.

Minor points:

1. With low contrast and crowded space between two membranes, how were Env-CD4 complexes at membrane-membrane interfaces picked manually? No sufficient details were given for Subtomogram averaging. How was an initial reference generated for alignment?

Response: We collected cryo-ET data with a Volta Phase Plate (VPP), which significantly increases the contrast of the images (Radostin, Danev *et al.*, PMID: 28109158). Although Env trimers were crowded at the interfaces, the Env-CD4 receptor complexes were unambiguous and we could pick them manually. The images below show a representative tomogram with the original manually picked Env trimers at interfaces. There was no initial reference. The global average was generated from these picked particles. We then performed iterations of alignment using the classification procedure in Protomo/I3. Any particles selected repeatedly were removed in later iterations of alignment. We now present a gallery of individual images depicting Env bound to one, two or three CD4 receptor molecules to illustrate that these events are visible in the raw data (Extended Data Fig. 4b, c).

2. “Quantification of tomograms revealed that membrane-membrane interfaces with 156 immature capsids had fewer Env trimers (4.6 ± 2.5) than interfaces with mature capsids (7.5 ± 4)”. Is the difference between the range of 2.1-7.1 and 3.5-11.5 really significant?

Response: As we indicated in the figure legend, the p-value from an unpaired t-test between mature and immature particles was significantly different ($P < 0.05$), having a p-value of less than 0.0001. We are presenting the data in a violin plot in the revised manuscript for better evaluation and clarity (see below). As indicated in the plot below, mature particles had significantly more Env trimers at membrane-membrane interfaces than immature particles.

3. “There are considerable steric constraints in the gp120-CD4 interaction that must be overcome for the Env trimer to bind the co-receptor. This could be achieved by either 1) shedding of the gp120-CD4 complex”. Is coreceptor binding to dissociated gp120 relevant to membrane fusion?

Response: We appreciate the question. We don't believe that binding of shed gp120 to co-receptor has been observed. We have modified the text to say release of CD4 may facilitate the transition of gp120 to coreceptor.

Referee #3 (Remarks to the Author):

The study uses cryo-ET imaging and subtomogram averaging to investigate the interaction of HIV-1 Env and its receptor CD4 in the context of native membranes. Subtomogram averaging is used to determine low resolution structures of the Env trimer in complex with CD4 including one, two, and three CD4-bound states. The asymmetric one and two bound states have not been observed before and are associated with an opening of the trimer based on comparison to previously described closed (unliganded) and open (3 bound) structures. Subsequent co-receptor engagement and membrane fusion requires additional structural changes.

This work describes a novel and well-designed system to observe interactions between Env and CD4 on the membranes of HIV-1 BaL particles and MLV-CD4 VLPs which cluster in the membrane at the interface between the two types of particles. Using cryo-EM imaging, the ms reveals structural changes that occur when Env interacts with CD4 receptors. Clustering is dependent on the maturation state of HIV and is shown to depend on free mobility of cytoplasmic tails. When the co-receptor CCR5 is added along with CD4, membrane fusion occurs, though CCR5 is not part of the structural investigation and the number of trimers involved in membrane fusion is not specifically determined.

The images and structures are extremely interesting and the observation of asymmetric binding and opening of Env and the mode of CD4 interaction with the membrane are important for understanding HIV entry and to extend structural work mostly based on isolated proteins. The main concern is the low resolution of the structures which limits interpretation. Re-arrangement of specific Env structural elements is plausible but not definitive at the level of protomer structure. The one CD4-bound structure has very weak density. A larger dataset by the same approach is desirable and would lead to more robust conclusions and impact.

Response: This work represents a trade-off between biology of highly heterogenic systems and resolution. HIV-1 Env trimers engaging CD4 receptors in biological membranes is a highly heterogenic system. Many factors contribute to this including the dynamics of Env, membrane-embedded proteins, and full-length CD4 having a higher degree of freedom than soluble trimers and soluble ligands (just D1D2 domains for CD4). While increasing the dataset may improve resolution to some extent, its effectiveness largely depends on the homogeneity of the samples. We are concerned that increasing the dataset for this highly dynamic system would only introduce more subclasses of the complexes with trivial improvement of the resolution. Thus, the question for us has been, what conclusions can we draw that are based on sound reasoning while accepting that only low resolution can currently be achieved? In our view, our data support two main points; 1) as HIV-1 Env approaches membranes, it binds one and two CD4 receptor molecules before it binds three, and 2) individual protomers reside in distinct conformational states in Env trimers with one and two bound CD4 molecules. This can be concluded due to characteristic density for the outwardly projecting V1V2 loops in the CD4-bound conformation and the lack of this density in protomers not bound to CD4. We believe that these are sound conclusions.

In an unexpected development, high-resolution structures have been provided by the independent report submitted in parallel by Dam *et al.*, who solved the structures of engineered asymmetric soluble trimers that can only bind one or two CD4 molecules. The soluble Env trimer bound to two D1D2 CD4 molecules is consistent with the density that we see in our data with Env in biological membranes. In this asymmetric trimer, the protomer that does not bind CD4 has the V1V2 loops still at the trimer apex consistent with the absence of the density for the outward projecting V1V2 loops in our cryo-ET density.

Initially the Env trimer bound to one CD4 molecule was an open trimer in our system, but a closed trimer in the Dam *et al.* report. However, upon careful re-analysis, Dam *et al.* identified the presence of an open trimer with one CD4 molecule bound that fits well into the cryo-ET density of our Env trimer bound to a single CD4 molecule. Dam *et al.* have included this new analysis in their revised manuscript. Thus, the two manuscripts are even more complementary, particularly as it is known that the Tier 2 HIV-1 isolate BG505 used for SP cryo-EM is more CD4-resistant, while the Tier 1b BaL isolate used for the cryoET studies is known to be more responsive to CD4. Thus, the two manuscripts support each other. CryoEM applied by Dam *et al.* reaches high resolution with molecular insights into the trimer structure but has to rely on engineered asymmetric trimers without knowing if they exist in biological membranes. Our report

documents that these asymmetric trimers do exist in biological membranes. Together, both reports represent an analysis of how HIV-1 Env engages CD4 receptor across multiple scales of resolution.

Specific points:

1. Quantitative assessment of opening or asymmetry should be included and is important in describing the findings.

Response: We appreciate this suggestion. We have quantified and presented a portion of the data in Extended Data Fig. 5 and Extended Data Table 1, and we also provide measures for openness and asymmetry in Figure 4 of revised manuscript as well as in the text.

2. The subtomogram averages appear to include specific non-parallel membranes associated with the interacting molecules, features which may be an important feature of the binding modes in the different types of particle interfaces and requires further analysis and discussion. This is important both to the data analysis and to comparison to other types of fusion systems such as SNAREs.

Response: We think the reviewer is referring to the membrane associated with the tilted HIV-1_{BaL} Env in the consensus averaging structure. We believe this comes from the tilting of Env on the HIV-1 surface. Subtomogram averaging the structure of Env trimer bound to one CD4 molecule also showed a more tilted HIV-1 membrane. We agree with the reviewer that this may be an important feature of the binding modes and performed the analysis of the distribution of the various Env structures at the interfaces. The results confirmed that Env trimers bound to two or three CD4 molecules were enriched in the region closer to the center of the interface (between parallel membranes), whereas Env trimers bound to one CD4 were distributed towards the periphery where the membrane curvature is increased. We include this analysis in Figure 3o of the revised manuscript.

3. Calculating subtomogram averages for unliganded data may be a valuable control that would demonstrate the quality of averaging and features obtained from more particles and confirm that the unliganded structures are as expected. This should be feasible given that the authors have already identified positions of those particles in their inter-molecule distance measurements.

Response: We appreciate the reviewer's suggestion to solve the cryo-ET structure of free, unliganded Env structures from the same data set. We performed this analysis for a similar ~6000 trimers in the same data set without imposing C3 symmetry and obtained the structure of a closed Env with a resolution of ~1 nanometer. Following symmetry expansion, we reached sub-nanometer resolution. This direct comparison within the same data set highlights that heterogeneity in the CD4-bound intermediates limits resolution. We include the structure of unliganded Env from the same data set in the revised manuscript (Extended Data Fig. 3e and 5d, h).

We also applied symmetry expansion on the Env trimer bound to three CD4 molecules. The binding angles of CD4 are similar as seen for the C1 structure (see below), indicating similar structural findings.

4. The data processing strategy for the tomography dataset involved classifying the subtomograms based on chosen membrane-membrane distances (method not described in sufficient detail) though the structures may in fact vary continuously with distance. A less biased approach would involve classifying all the subtomograms based on CD4 binding state and then identify their locations in the tomograms. They could also assign the number of different CD4 bound states for the different interaction clusters (loosely defined as small, large and rings in the ms). At present there are relatively few particles for the different categories and all initial chosen particles are retained in the final averages, without rejections based on quality, so improvements are possible.

Response: We thank the reviewer for this suggestion. We actually performed subtomogram classification based on CD4 binding. The result were similar to classifying subtomograms based on membrane-membrane distances (see figure below). The heat map of the Env positions for one, two and three CD4 receptor bound trimers (Fig. 3o of revised manuscript) explains why, because the Env trimers with one CD4 molecule bound are located in the periphery where the distance between membranes is greater. In contrast, Env bound to two and three CD4 molecules were positioned closer to the center of the interfaces, resulting in shorter membrane-membrane distances. This explains why both classification strategies arrive at similar results.

Referee #4 (Remarks to the Author):

Review for Li et al. MTS-2022-12-20465A

The authors present a concise manuscript outlining the acquisition of structural information of HIV-1's Env complexes in viral context; a comparative analysis to MLV's Env is also derived as part of the present work. The experiments carried out in the present manuscript are certainly of interest to a broad community as there exists a debate about the structural details of HIV Env in infective particles. The work performed by the authors is in the context of infectious particles which positions them in a unique situation compared to previous work. However, the lack of sub-nanometer structures and systematic analysis of the derived models reduces the enthusiasm of this reviewer.

Major concerns

1. The major finding from the manuscript seems the asymmetric nature of Env trimers when bound to CD4 receptors. However the lack of sub nanometer structures in the present work limits the interpretability of the structural data. Furthermore the findings are not supported by virological assays.

Response: We understand that reviewers 3 and 4 are asking for structures with higher resolutions. As we explain above to Reviewer 3, we can solve a sub-nanometer structure of the closed Env trimer for the trimers outside of membrane-membrane interfaces indicating that the low resolution originates from heterogeneity, not from the data set itself. It is also Pamela Bjorkman's experience that even at the level of soluble Env trimers, CD4-bound trimers are always of lower resolution than closed, unliganded trimers, likely because of their higher heterogeneity (*i.e.*, asymmetry). Because increasing resolution is very challenging in highly heterogenic systems using current cryo-ET technology, we thus focused on the biology of a dynamic process (see the general response to Reviewer 3). Importantly, our conclusions on HIV-1 Env binding to one CD4 molecules, two CD4 molecules, and finally three CD4 molecules are solid conclusions that can be made with the current resolution. Similarly, the outward movement of the V1V2 loops (and their absence) is clearly visible at the current resolution.

We are also encouraged by the parallel high-resolution of soluble engineered asymmetric Env trimers bound to one and two CD4 molecules submitted by Dam *et al.* as they are in general agreement with the intermediates seen in our data sets in biological membranes. The structure of the open trimer bound to two CD4 receptor molecules is consistent with the density observed in our membrane-membrane interfaces. The main structure of the trimer bound to one CD4 molecule was a closed trimer for Dam *et al.*, but an open trimer in our biological membranes. Interestingly, a careful re-analysis by Dam *et al.* revealed that one third of the BG505 HT1 trimers bound to CD4 are open trimers with mostly one CD4 bound. The revised manuscript by Dam *et al.* includes this additional structure. The open trimer bound to one CD4 molecule fits well into the cryo-ET map of our trimer with one bound CD4 molecule. The two manuscripts are thus quite consistent, particularly as it is expected that a more CD4-resistant Tier 2 HIV-1 isolate such as BG505 would be less responsive to CD4 than the Tier 1b isolate BaL used in our cryo-ET work.

We had developed and performed split nanoluciferase complementation assays to investigate fusion kinetics as a more virological assay. The assay results align with the findings obtained from the cryoET work. However, as mentioned by Reviewer 1, further investigation is needed to explore the significance of Env clustering in fusion. Therefore, we have decided to exclude the fusion kinetics analysis from the manuscript and instead concentrate on the structural findings.

2. The authors refer to their models as atomistic; this reviewer begs to differ. No attempt was made to assess the quality of the MDFF-derived models via any standard metrics. For instance, there is not information regarding the structural stability of the resulting models (*e.g.*, root-mean-squared differences traces, model-to-map fits, etc). There is not assessment on the cross correlation between model and density. Certainly, the maps could have been used in a more rigorous way to produce higher quality models.

Response: We acknowledge the reviewer's concerns, provide the MDFF trajectories, CCC fits, and more details on MDFF in the revised manuscript. We also change the language to avoid the use of atomistic models. Our intention was to present the global changes in structure and openness of these trimers, not atomic details of the MDFF models. The MDFF runs indicate that the partially open 6CM3 structures explain the cryoET density better than the more open 5VN3. Note that the sequence of the HIV-1 Env in both structures differs. 5VN3 was performed for Env of B41 isolate, 6CM3 for BG505. To compensate for the different HIV-1 isolates, we generated homology models of these structures for BG505 and B41, respectively, and run MDFF for all structures resulting in similar results. These data are presented in Figure 4 and Extended Data Figure 6.

2. The analysis pertaining to the models derived from the densities is ambiguous and not very rigorous (see point #2). The authors refer to an open, closed and "more open" conformation. The ambiguity is not

only in the language, but also the authors should define unbiased metrics that allow them to classify the models unambiguously.

Response: We thank the reviewer for alerting us to the ambiguity of our language. The extended data Table 1 contains all data derived from the MDFF modeling, including all distances that define degrees of openness. We also provide a revised Figure 4 to present more details about degrees of openness.

3. As with any manual selection of VLPs there is the possibility of introducing a human-bias. Although the authors are experts in the field of cryoET, it is the view of this reviewers that there should be an attempt to convince readers that the results are invariant to particle selection.

Response: We appreciate the reviewer discussing the possibility of introducing human bias. We have indicated how Env-CD4 complexes were manually picked in the response to Reviewer 2 minor point 1. The raw images have very high contrast due to the use of the VPP leading to unambiguous structures of Env bound to CD4. Because the structures are quite obvious in the raw images we are confident that our particle picking has not introduced human bias. Furthermore, the structure of free Env trimers solved from the same dataset (requested by Reviewer 3) showed a closed Env, suggesting that the averaged structures are not affected by reference bias.

Minor concerns

1. The MDFF simulations calculations were performed 'in vacuo'; these types of calculations are acceptable for specialized journals. At the level of Nature, this reviewer would expect that the derivation of full-length models in realistic lipid membranes would be the norm.

Response: We would like to emphasize that our use of MDFF in this work was not meant to create a full-atom simulation of Env behavior in membranes. Instead, we intended to use MDFF as a tool to investigate the higher-level biology of HIV-1 Env binding to CD4. In this limited analysis, we found that the published partially open Env atomic model fits our cryo-ET density better than the fully open Env atomic model. We believe this is a sound conclusion given the high-level view of our structures and our limited use of the MDFF method.

Reviewer Reports on the First Revision:

Referees' comments:

Referee #1 (Remarks to the Author):

The authors have answered all my questions (sometimes supporting their arguments with data) and have modified the text accordingly.

They have also focused on structural parameters regarding the step-wise interactions between Env and CD4(n) molecules and their claims regarding the role of clusters in fusion are now less stringent.

I am happy with the explanations and the experiments supporting the author's claims and therefore, in recommend this article for publication in Nature.

Referee #2 (Remarks to the Author):

The authors have addressed some of my previous concerns, but lack of rigorous characterization of the virus sample remains an issue. A western blot has been provided to show that the Env trimers are fully cleaved when produced in SupT1 cells, which is helpful (There is something odd about the western blot though. It is claimed that fibroblast cells produce mostly bald HIV-1 particles with no Env trimers on their surfaces, but the western blot shows a similar level of gp160+gp120 when compared to that of gp120 from the SupT1 cell-produced particles, even after roughly normalized by the p24 level). HIV-1BaL is a tier 1b virus, which in principle has less stable Env trimers. Again, does CD4 binding induce gp120 dissociation or not? A simple experiment by mixing the virus preparation with soluble CD4 should give the answer.

Referee #3 (Remarks to the Author):

Using a system to observe interactions between HIV-1 Env on virus particles and CD4 on VLPs for subtomogram averaging, I think the ms successfully demonstrates that states with 1, 2, and 3 CD4's bound can be observed in the context of membranes and this is a novel finding. I feel this demonstration has been improved in the revision by the calculation of the unliganded map control, further explanation of the data processing, and the additional information on the location of different CD4-bound species within the clusters and rings at the interfaces. The heat maps (Figure 3o) and the new Extended Data Figure 4 are helpful.

The information content of the maps appears sufficient to show whether CD4 is bound and that the trimers have opened-up. The maps may be consistent with interpretation of medium resolution protomer features (e.g. where are the V1V2 loops) but are not of sufficient detail to determine these independently. This is the major limitation of the study and therefore does not alone provide a specific structural mechanism which is important to understand the role of these complexes as structural intermediates.

Since there are relatively few particles used for averaging across several different species (~5700), more data for CD4-bound states would improve the robustness of the data generally and modest resolution improvement would increase significance by confirming interpretation of structural features of individual protomers. The idea that more data would just produce more subclasses may be a possible outcome and would be interesting but is not in line with the main interpretation of the current dataset that there are asymmetric CD4-bound intermediates with protomers in distinct states. This would provide stronger validation or refutation of single particle studies of prepared asymmetric complexes with the advantage of seeing all native states on membranes.

Specific points:

1. The first round of review included a table on refinement and validation statistics which I did not find as part of the revision. It is important to include such a table, including the additional states in the revision (e.g. unliganded) and where models are described in the text and figures, including map-model agreement statistics (e.g. map-model FSC) and internal geometry which are standardly reported for fitted models. Anisotropy of map data should be reported if it is present in a small dataset.
2. Lines 137-138 indicate that numbers of Env trimers “increased from small clusters to larger clusters and rings”, but Figure 2e suggests that rings have fewer trimers than large clusters, so this should be clarified. As far as I can tell, the data do not test a progression from clusters to rings (line 303).

Referee #4 (Remarks to the Author):

The authors have addressed all my concerns and I look forward to seeing the manuscript published.

Author Rebuttals to First Revision:

Response to Reviewers

Referees' comments:

Referee #1 (Remarks to the Author):

The authors have answered all my questions (sometimes supporting their arguments with data) and have modified the text accordingly.

They have also focused on structural parameters regarding the step-wise interactions between Env and CD4(n) molecules and their claims regarding the role of clusters in fusion are now less stringent.

I am happy with the explanations and the experiments supporting the author's claims and therefore, in recommend this article for publication in Nature.

Response: We are glad that the reviewer finds that the revised manuscript is responsive to his/her previous concerns.

Referee #2 (Remarks to the Author):

The authors have addressed some of my previous concerns, but lack of rigorous characterization of the virus sample remains an issue. A western blot has been provided to show that the Env trimers are fully cleaved when produced in SupT1 cells, which is helpful (There is something odd about the western blot though. It is claimed that fibroblast cells produce mostly bald HIV-1 particles with no Env trimers on their surfaces, but the western blot shows a similar level of gp160+gp120 when compared to that of gp120 from the SupT1 cell-produced particles, even after roughly normalized by the p24 level). HIV-1BaL is a tier 1b virus, which in principle has less stable Env trimers. Again, does CD4 binding induce gp120 dissociation or not? A simple experiment by mixing the virus preparation with soluble CD4 should give the answer.

Response: We apologize for this miscommunication on our side.

(REDACTED)

While it often appears that Env is released into the supernatant for HIV-1NL4-3 produced from HEK 293, Env incorporation into particles is very poor. Occasionally, we could observe a few viruses that incorporated some Env trimers on the particles (red arrow), but most particles are bald. The inability to generate HIV-1 particles of high quality is the main reason why cryoET has relied on HIV-1BaL produced from chronically infected SupT1 cells.

With respect to shedding of gp120, we previously reported gp120 shedding for BaL Env by soluble CD4 and 17b Fab in Li et al., 2020 (PMID: 32601441). As can be seen from Table 1 presented below, we had to record ~3 times more tomograms in the presence of sCD4 and 17b to arrive at a similar number of Env particles as other samples because particles presented much less Env on the virion surfaces. This data indicates that sCD4 and 17b induce shedding of gp120 on BaL virions observed by EM.

Table 1 | Cryo-ET data collection, refinement and resolution estimation

	HIV-1 _{BAL} Env ligand-free (EMD- 21412)	HIV-1 _{BAL} Env- CD4-17b (EMD-21411)	HIV-1 _{BAL} Env-10-1074- 3BNC117 (EMD-21413)
Data collection and processing			
Magnification	64,000	64,000	64,000
Voltage (kV)	300	300	300
Electron exposure (e ⁻ /Å ²)	50	50	50
Defocus range (μm)	0 (VPP)	0 (VPP)	0 (VPP)
Pixel size (Å)	2.2	2.2	2.2
Symmetry imposed	C ₃	C ₃	C ₃
Tomograms (no.)	70	208	85
Virions (no.)	683	959	713
Initial particle images (no.)	13,577	13,817	15,243
Final particle images (no.)	12,219	12,435	13,718
Map resolution (Å)	11.5	9.7	9.9
FSC threshold	0.5	0.5	0.5
Map resolution range (Å)	7-14.5	7-12	7-11

Upon request of the reviewer, we also performed a gp120 shedding-experiment by incubating HIV-1_{BAL} viruses with sCD4 at room temperature, subsequently sedimenting viruses to ask how much gp120 was released into the supernatant (see below). The result showed that some gp120 shedding was triggered by sCD4 binding in both, the presence and absence of 17b. In the current manuscript, we focused on the Env bound to full-length CD4 molecules at the interface between HIV-1 and MLV particles. The structure of Env bound to three CD4 molecules on membranes determined by cryoET shows Env in a partially open state. We believe that HIV-1 evolved these partially open conformational states when bound to CD4 receptor on membranes exactly to prevent shedding. We hypothesize that in biological membranes, unstable intermediates are only transiently exposed when Env can already contact coreceptor. In

agreement, no gp120 is released into the supernatant when HIV-1_{BaL} particles are incubated with MLV-CD4 particles, and no loss of trimers is observed at the cryoET level. We include this data now in Extended Data Figure 1 and discuss it in the text (lines 105-108 and 296-304).

sCD4 and sCD4 + 17b induce gp120 shedding on HIV-1_{BaL} virions. HIV-1_{BaL} viruses were incubated at room temperature for 1 hour with PBS, sCD4 (100 ug/ml) alone, sCD4 (100 ug/ml) in combination with 17b Fab (100 ug/ml), or MLV-CD4 VLPs. Following incubation, the samples were subjected to ultracentrifugation at 130,000 g through a 20% sucrose cushion for 1 hour to separate the supernatant (S) and pellet (P) fractions. Western blotting was performed using 2G12 antibody to detect gp120 and anti-HIV-1 serum to detect gp41 and p24. Residual 17b Fc fragments in 17b Fab were detected by anti-Human 2nd HRP-antibody (grey).

Referee #3 (Remarks to the Author):

Using a system to observe interactions between HIV-1 Env on virus particles and CD4 on VLPs for subtomogram averaging, I think the ms successfully demonstrates that states with 1, 2, and 3 CD4's bound can be observed in the context of membranes and this is a novel finding. I feel this demonstration has been improved in the revision by the calculation of the unliganded map control, further explanation of the data processing, and the additional information on the location of different CD4-bound species within the clusters and rings at the interfaces. The heat maps (Figure 3o) and the new Extended Data Figure 4 are helpful.

The information content of the maps appears sufficient to show whether CD4 is bound and that the trimers have opened-up. The maps may be consistent with interpretation of medium resolution protomer features (e.g. where are the V1V2 loops) ...

Response: We agree with the reviewer that these are the main findings of our manuscript. HIV-1 Env binding steps to 1, 2, and 3 CD4 receptor molecules can be readily observed in membranes suggesting that they are intermediates in the binding of viruses to cells. The resolution is also sufficient to conclude that these trimers have opened up. Moreover, because the outwardly moving V1V2 loops are a characteristic feature of the CD4 bound protomers even at the resolution achieved by cryoET, their absence in CD4-unbound protomers indicates that the protomers must be in distinct conformations. The trimers are therefore asymmetric. As we will outline in detail below, beyond these main conclusions, our manuscript can't provide details about where the V1V2 loops reside in the unbound protomers. Rather we have to rely on atomic models provided by the parallel cryoEM structures of soluble trimers bound to 1, and 2 CD4 molecules that are in good overall agreement. Both groups have presented back-to-back at several meetings recently (including the NIH HIV structural biology meeting in Bethesda) and the overall agreement between the two approaches was appreciated by the audience.

... but are not of sufficient detail to determine these independently. This is the major limitation of the study and therefore does not alone provide a specific structural mechanism which is important to understand the role of these complexes as structural intermediates.

Since there are relatively few particles used for averaging across several different species (~5700), more data for CD4-bound states would improve the robustness of the data generally and modest resolution improvement would increase significance by confirming interpretation of structural features of individual protomers. The idea that more data would just produce more subclasses may be a possible outcome and would be interesting but is not in line with the main interpretation of the current dataset that there are asymmetric CD4-bound intermediates with protomers in distinct states. This would provide stronger validation or refutation of single particle studies of prepared asymmetric complexes with the advantage of seeing all native states on membranes.

Response: We understand. What we likely have to do here is to explicitly state these limitations (lines 333-335). Many journals now have a paragraph devoted to the limitation of the study where these

questions are addressed with transparency. Importantly, we believe that the expectations that the apex can be resolved by cryoET, expressed here by the reviewer, come from cryoEM single particle analysis (SPA) field, which is influenced by the unusually stable trimers of the BG505 isolate that are further stabilized by the SOSIP mutations. It took the Sanders/Moore/Ward groups 10-15 years to 1) identify a highly stable Env screening numerous HIV-1 isolates, and 2) introduce additional stabilizing mutations (SOSIP) that generate stable trimers enabling high-resolution structures (PMIDs: 10623724; 10799583; 12097589; 12163607; 24068931). In contrast, the apex of many transmitted/founder viruses is more flexible (Priyamvada Acharya and Alon Herschhorn, personal communication). Moreover, HDX-MS data generated by Kelly Lee working with Rogier Sanders documents that in contrast to BG505, the trimer apex formed by the V1V2 and V3 loops is highly flexible for the Tier 1b isolate AMC008 (PMID: 35677643). Similarly, the BaL viruses used here for cryoET are a tier 1b isolate. Thus, even if we would reach ~5-6 Å resolution, we do not expect to be able to resolve density for the apex in the unbound protomers of the trimers bound to one and two CD4 molecules.

Increasing the dataset size is a common solution to increase resolution in cryoEM SPA. However, cryoET data collection and analysis are much less efficient and more labor-intensive compared to SPA. Occasionally, high-resolution structures have been determined using cryoET, particularly when the target proteins exhibit high homogeneity, as observed with HIV-1 capsid. In our case, we are dealing with high heterogeneity. The critical sub classes in need of higher resolution are Env bound to one CD4 molecules (currently ~2000 particles), and Env bound to two CD4 molecules (currently ~1600 particles). Reaching ~10,000 particles in each sub class would require a 5-, or 7- fold larger data set. It would require ~6 months of work and still constituting only 5%-10% of the numbers achieved to reach high resolution at the SPA level (Dam, Fan and Bjorkman). To achieve the high resolution, Dam, Fan et al. had to reduce the heterogeneity of CD4-bound trimers by artificially engineering trimers that can only bind one or two CD4 molecules. Priyamvada Acharya needs above 3 million particles to resolve structural heterogeneity in the apex of a non-BG505 isolate. This is very hard to achieve at the cryoEM SPA level, and currently not possible at the cryoET level.

Instead of increasing the dataset, we tested what happens when we reduced the number of particles by randomly extracting half of our dataset and determined the resolution of the averaged structures. As shown below, the resolution approximately decreased 3-4 Å when compared to the structures from the full-size dataset (from blueish to greenish), but without losing any information regarding to the number of bound CD4 molecules and the density of V1V2 loops. This implies that increasing the dataset would provide limited additional information to our structures due to diminishing returns in resolution improvement as the dataset increases further. As outlined above, even if we were to achieve 5-6 Å resolution, the apex of the Tier 1b BaL Env trimers is predicted to stay unresolved.

We believe that both manuscripts have their limitations, yet complement each other and are therefore stronger together. Dam, Fan et al. artificially engineered trimers that only bind 1 or 2 CD4 molecules, which allows them to reduce the heterogeneity of Env-CD4 complexes to arrive at structures high enough to present atomic resolution models for the conformational state of Env in these asymmetric trimers. However, whether they exist under physiological conditions on membranes is unknown. In our work, we visualized how HIV-1 viruses interact with CD4 receptor embedded in membranes to readily detect asymmetric trimers bound to 1, 2 and 3 CD4 receptor molecules. Because we can see the density for the outwardly projecting V1V2 loops in CD4-bound protomers, their absence in unbound protomers indicates the existing of asymmetry. Moreover, the Dam, Fan et al. structures of the open BG505 Env bound 1 CD4 and 2 CD4 molecules fit well into our cryoET density. The observation of an additional closed BG505 trimer bound to a single CD4, and the unresolved density in the apex of HIV-1_{BaL} particles at the cryoET level are an expected outcome as BG505 is a highly CD4-resistant HIV-1 isolate, and BaL as tier 1b known to have a variable apex. The two manuscripts are such in good agreement and present structural studies of HIV-1 Env – CD4 receptor interactions across scales of 3 – 20 Å resolution.

Specific points:

1. The first round of review included a table on refinement and validation statistics which I did not find as part of the revision. It is important to include such a table, including the additional states in the revision (e.g. unliganded) and where models are described in the text and figures, including map-model agreement statistics (e.g. map-model FSC) and internal geometry which are standardly reported for fitted models. Anisotropy of map data should be reported if it is present in a small dataset.

Response: We apologize that we missed the table in revision. The table was provided by Nature for data collection, refinement and validation. In our case, we could only report the data collection. Since the resolution of maps were modest, we did not perform refinement for the models.

	Unliganded HIV-1 Env (EMDB- 41045)	HIV-1 Env bound to one CD4 (EMDB- 29292)	HIV-1 Env bound to two CD4 (EMDB- 29293)	HIV-1 Env bound to three CD4 (EMDB- 29294)	Consensus structure of HIV-1 Env bound to CD4 (EMDB- 29295)
Data collection and processing					
Magnification	300	300	300	300	300
Voltage (kV)					
Electron exposure (e-/Å ²)	123	123	123	123	123
Defocus range (µm)	-0.5	-0.5	-0.5	-0.5	-0.5
Pixel size (Å)	2.7	2.7	2.7	2.7	2.7
Symmetry imposed	C3	C1	C1	C1	C1
Initial particle images (no.)	6444	5712	5712	5712	5712
Final particle images (no.)	6339	2341	1418	1953	5712
Map resolution (Å)	9	17	19	16	15
FSC threshold	0.5 cutoff	0.5 cutoff	0.5 cutoff	0.5 cutoff	0.5 cutoff
Map resolution range (Å)					

The map-model agreement statistics and internal geometry recommended by the reviewer are typically employed in cryo-EM single particle analysis when the map resolution is superior to 4 Å. The reliability of the models increases significantly when they are built based on higher resolution maps. Since our cryo-electron tomography (cryo-ET) map does not possess high resolution, the analysis of the atomic model was focused on the overall domain rather than atomic-level details. The models will not be deposited as complete atomic structures. These are again expectations originating from the cryoEM field that are not helpful with resolutions achieved by cryoET.

Nonetheless, in order to validate the fitted model obtained through MDFF, we performed MDFF simulations starting from different structures and compared the backbone-RMSD of the final frames. Additionally, we have also shown the cross-correlation coefficient (CCC) curve between the density map and the trajectories of 5VN3 and 6CM3 MDFF simulations, which depicts the quality of the fitting. As the reviewer request, the map-model Fourier shell correlation (FSC) is provided below: The curves derived from fitted models, 5VN3-CD4 and 6CM3-CD4, were similar. The resolution matched the low-pass filtered (20 Å) density map used for MDFF.

During MDFF, we employed additional elastic restraints for internal geometry elements, such as secondary structure dihedrals, hydrogen bonds, chiral centers, and cis-peptide bonds, to preserve the structural integrity of the secondary elements and prevent overfitting. Ramachandran plots are provided below.

Regarding anisotropy, we analyzed the particle orientations and plotted their distribution. As shown here, the unliganded Env particles indicate an even distribution across all orientations. However, the Env particles bound to CD4 molecules showed reduced top-view and bottom-view particles. This is because the Env-CD4 complexes were picked at the interface between two viruses, resulting in a higher occurrence of side-view orientations. The top-view and bottom-view of the interfaces would be too thick to be captured by cryoET. This is another limitation in obtaining high-resolution structures of Env-CD4 complexes on membranes. It is worth noting that no orientation preference was observed among the Env bound to one, two and three CD4 molecules. We have updated this analysis in Extended Data Fig. 4.

2. Lines 137-138 indicate that numbers of Env trimers “increased from small clusters to larger clusters and rings”, but Figure 2e suggests that rings have fewer trimers than large clusters, so this should be clarified. As far as I can tell, the data do not test a progression from clusters to rings (line 303).

Response: Thanks. We corrected the text accordingly (line 141). We acknowledge that our data does not support the progression from clusters to rings and have corrected the text accordingly (line 315).

Referee #4 (Remarks to the Author):

The authors have addressed all my concerns and I look forward to seeing the manuscript published.

Response: We are pleased that the revised manuscript addresses the reviewer’s previous concerns.

Reviewer Reports on the Second Revision:

Referees' comments:

Referee #2 (Remarks to the Author):

The authors have addressed my concerns, although the virus sample could have been characterized more using broadly neutralizing antibodies to ensure conformational homogeneity required for averaging in cryo-EM analysis.

Referee #3 (Remarks to the Author):

The map-model FSC is the best way to report overall map-model agreement, even at low resolution. The only remaining request on this ms is that the map-model FSC curves presented in the rebuttal should be included in Extended Data. Perhaps in Extended Data 6.

Just a comment: According to the author responses on this point, it is possible that the models from the companion study of Dam, Fan et al. will have a better agreement.

Author Rebuttals to Second Revision:

Response to Reviewers and to the Editor

We are delighted that reviewers believe that we have been responsive to their concerns and suggestions. As suggested by reviewer 3, we present an improved map-model FSC as panel h in Extended Data Fig. 7. We also tested if the new atomic structures by Dam et al. fit better into the density of the cryoET density of HIV-1BaL Env bound to 3 CD4 molecules. It turns out that they do not and 6CM3 remains the best fit for the observed partially open Env trimer. We therefore do not include additional MDFF data.

We adhered to the instructions to cut main text by ~650 words, keep title under 75 characters with spaces, all subtitles under 40 characters with spaces, and figure legends under 300 words.

We also updated the statistic analysis section in Method. Statistical significance for pairwise comparisons were derived by applying non-parametric Mann-Whitney test (two-tailed), Extended Data Figure 2b,d. To assess statistical significance for multiple comparisons we employed Kruskal-Wallis test followed by Dunn's multiple comparison tests, Figure 2e, f, and Extended Data Figure 3d. p values lower than 0.05 were considered statistically significant. P values were indicated as *, $p < 0.05$; **, $p < 0.01$; ***, $p < 0.001$; ****, $p < 0.0001$.

Referee #2 (Remarks to the Author):

The authors have addressed my concerns, although the virus sample could have been characterized more using broadly neutralizing antibodies to ensure conformational homogeneity required for averaging in cryo-EM analysis.

Response: We are glad that we addressed the reviewer's concerns and appreciate the suggestion.

Referee #3 (Remarks to the Author):

The map-model FSC is the best way to report overall map-model agreement, even at low resolution. The only remaining request on this ms is that the map-model FSC curves presented in the rebuttal should be included in Extended Data. Perhaps in Extended Data 6.

Just a comment: According to the author responses on this point, it is possible that the models from the companion study of Dam, Fan et al. will have a better agreement.

Response: We have included a map-model FSC that was generated using a 15 Å low-pass filtered density map, with the membranes appropriately masked out. This curve has been added to the revised manuscript as panel h of Extended Data Fig. 7.

In response to the reviewer's suggestion, we conducted MDFF simulations using the models from Dam, Fan et al.'s study for Env bound to two CD4 and to three CD4 molecules. The results indicated no substantial differences for Env bound to two CD4 molecules, consistent with their observations. The model 6CM3 remains the best fit for Env bound to three CD4 molecules. Given these findings, we will not make further revisions or include additional analyses to our manuscript.